# Unraveling the Genetic Landscape of Neurological Disorders: Insights into Pathogenesis, Techniques for Variant Identification, and Therapeutic Approaches

**DOI:** 10.3390/ijms25042320

**Published:** 2024-02-15

**Authors:** Zeba Firdaus, Xiaogang Li

**Affiliations:** 1Department of Internal Medicine, Mayo Clinic, Rochester, MN 55905, USA; firdaus.zeba@mayo.edu; 2Department of Biochemistry and Molecular Biology, Mayo Clinic, Rochester, MN 55905, USA

**Keywords:** next generation sequencing, gene therapy, Alzheimer’s disease, Parkinson’s disease, amyotrophic lateral sclerosis

## Abstract

Genetic abnormalities play a crucial role in the development of neurodegenerative disorders (NDDs). Genetic exploration has indeed contributed to unraveling the molecular complexities responsible for the etiology and progression of various NDDs. The intricate nature of rare and common variants in NDDs contributes to a limited understanding of the genetic risk factors associated with them. Advancements in next-generation sequencing have made whole-genome sequencing and whole-exome sequencing possible, allowing the identification of rare variants with substantial effects, and improving the understanding of both Mendelian and complex neurological conditions. The resurgence of gene therapy holds the promise of targeting the etiology of diseases and ensuring a sustained correction. This approach is particularly enticing for neurodegenerative diseases, where traditional pharmacological methods have fallen short. In the context of our exploration of the genetic epidemiology of the three most prevalent NDDs—amyotrophic lateral sclerosis, Alzheimer’s disease, and Parkinson’s disease, our primary goal is to underscore the progress made in the development of next-generation sequencing. This progress aims to enhance our understanding of the disease mechanisms and explore gene-based therapies for NDDs. Throughout this review, we focus on genetic variations, methodologies for their identification, the associated pathophysiology, and the promising potential of gene therapy. Ultimately, our objective is to provide a comprehensive and forward-looking perspective on the emerging research arena of NDDs.

## 1. Introduction

With an increase in the aging population, neurodegenerative disorders (NDDs) are among the major health issues in the modern world. It is estimated that they will become the second leading cause of death globally by 2050, surpassing cancer [1]. Neurodegenerative disorders that are chronic and progressive exhibit a distinct pattern of neuron loss in motor, sensory, or cognitive systems [2]. The common symptoms of NDDs include impairment of the motor system, sensory network, cognitive function, memory, and abstract thinking that could appear during the disease progression [3]. Alzheimer’s disease (AD), Parkinson’s disease (PD), and amyotrophic lateral sclerosis (ALS) take the forefront among neurodegenerative disorders, resulting in the unfortunate demise of thousands of Americans. Moreover, the patient count is anticipated to rise steadily in the upcoming decades [2,4].

NDDs primarily emerge during late adulthood and are frequently associated with the accumulation of protein aggregates. For instance, AD is predominantly characterized by the aggregation of amyloid β and tau proteins, while PD and ALS are characterized by the accumulation of α-synuclein [5] and TDP43 [6], respectively (Figure 1). While the initial diagnosis of an NDD is typically based on clinical presentation, definitive confirming requires post-mortem pathological analysis to identify the specific protein aggregates. Moreover, the presentation of NDDs varies widely, and it is increasingly recognized that the diagnosis exists on a spectrum, with a greater prevalence of mixed pathology and overlapping clinical features than previously acknowledged [5,7]. Understanding the underlying causes and pathological processes of neurological diseases stands as one of the most critical challenges in the fields of medical and biological sciences due to their relatively high prevalence, largely unknown mechanisms, and significant impact on affected individuals, families, and society.

NDDs are influenced by a wide range of genetic factors, from straightforward direct predisposition in diseases like Huntington’s disease and spinocerebellar atrophy to more complex roles in diseases like AD and PD [8,9]. A number of causative genes for familial forms of NDDs have been identified; these genes are inherited as Mendelian traits [10], and their discovery has advanced knowledge of the molecular mechanisms underlying the distinctive neuronal degeneration that distinguishes each disorder. Only a small percentage of ALS, AD, and PD cases (5–10%) are familial, while the vast majority (>90%) are sporadic [11], most likely due to complex interactions between genetic and environmental factors in addition to the slow, sustained neuronal dysfunction caused by aging.

This review is primarily dedicated to dissecting the complex web of genetic changes in neurological disorders such as ALS, AD, and PD. Our investigation extends to the methodologies employed for pinpointing these alterations. Through this review, we aspire to paint a broad and forward-looking canvas, capturing the dynamic landscape of research in neurodegenerative diseases.

## 2. Genetic Techniques in Addressing Neurodegenerative Disorders

In the realm of molecular genetics and genetic epidemiology, disease-related genes are typically classified into two principal groups: causative genes and susceptibility genes [12]. Through the identification of the causal or susceptibility genes for each neurological disorder, along with the utilization of transgenic techniques, our grasp of the fundamental molecular mechanism has improved, creating a pathway for the exploration of potential therapeutic targets. In this section, we will discuss different cutting-edge high throughput technologies to identify the genetic variation in neurodegenerative disorders (Table 1).

Many neurological disorders are known to have substantial genetic factors contributing to their development [13]. Recent progress in the fields of genome technologies, encompassing genome-wide association studies (GWASs) and next-generation sequencing (NGS) technology, has substantially enhanced our insights into the genetic factors contributing to these diseases [14]. In the case of rare Mendelian disorders, NGS exhibits the capacity to unveil novel genes harboring mutations that intricately underpin the phenotypic expressions [14]. GWAS, based on the common disease–common variant hypothesis, aims to elucidate how common genetic variability is associated with the development of prevalent diseases [15]. Most common neurological diseases like AD, PD, and ALS can be studied using both approaches. GWAS works well for sporadic cases (without a family history), while NGS-based studies are best for cases with a strong family connection, indicating a likely genetic inheritance [16].

Genome-wide association studies (GWASs) have successfully revealed numerous susceptibility genes for common diseases such as diabetes as well as NDD, the odds ratios associated with these risk alleles are generally low and account for only a small proportion of estimated heritability [17]. The theoretical framework for a GWAS is the ‘common disease—common variant hypothesis’, in which common diseases are attributable in part to allelic variants present in more than 1–5% of the population [18]. It is assumed that risk alleles with a large effect size may be rare in frequency and hard to detect with GWASs employing common single nucleotide polymorphisms (SNPs). The current experience with GWASs strongly suggests that rarer variants that are hard to detect with GWASs may account for the ‘missing’ heritability [19]. This suggests the presence of rare variants (found in less than 5% of the population) between the extremes of the frequency spectrum. Such rare variants may have large effect sizes as genetic risk factors for diseases. Thus, we need a paradigm shift from the ‘common disease—common variants hypothesis’ to a ‘common disease—multiple rare variants hypotheses’ to identify disease-relevant alleles with large effect sizes [20,21]. These rare variants, although not causative, typically have a significant impact, and GWASs using common SNPs struggle to capture them due to their large effect size [22].

The advancement in next generation sequencing technologies has notably enhanced the efficiency and affordability of both whole exome sequencing (WES) and whole genome sequencing (WGS) in the recent past [23]. In contrast to the extended timeframe and high expenses associated with first-generation sequencing, specifically Sanger sequencing which required several years and millions of dollars to sequence an entire diploid human genome, an NGS platform can achieve this sequencing within a few weeks, at a modest cost [24]. Furthermore, NGS technology has facilitated the discovery of rare variants with substantial effects, revealing missense or nonsense single-base substitutions, as well as small insertions or deletions. These findings bear significant implications for risk prediction, diagnosis, and the treatment of neurological diseases [14].

Despite the vast array of mutations identified through next-generation sequencing (NGS), its short-read lengths (150–300 bp) and limited representation in GC-rich/poor areas poses challenges in resolving expansions beyond several kilobases [25]. In comparison to NGS, two prevalent long-read sequencers referred as LRS, including single-molecule real-time (SMRT) sequencing developed by Pacific Biosciences (PacBio), Menlo Park, CA, USA [26] and nanopore sequencing by Oxford Nanopore Technologies (ONTs), Oxford Science Park, Oxford, UK [27], present an alternative method for sequencing single DNA molecules in real time (Figure 2). The reads generated by these platforms are exceptionally long, extending over several tens of kilobases and capturing entire repetitive regions [28]. Additionally, there is a potential to diminish the guanine-cytosine (GC) bias to a lesser degree and attain a more even genome coverage without the necessity for PCR amplification, unlike NGS [28,29]. In the realm of targeted sequencing, a noteworthy advancement is the integration of clustered regularly interspaced short palindromic repeats/CRISPR-associated 9 (CRISPR/Cas9)-mediated amplification-free enrichment with long-read sequencing (LRS), presenting an essential tool for targeted sequencing [30]. The distinctive attributes of LRS render it highly appropriate for unraveling neurodegenerative diseases, particularly in instances where NGS outcomes are uninformative.

In a more recent development within the human domain, the telomere-to-telomere consortium utilized long-read whole-genome sequencing (WGS) to successfully sequence the very first “complete human genome” [31,32]. The initiative commenced with the aim of constructing a human reference genome devoid of any gaps; thus, researchers utilized both PacBio and ONT technologies to meticulously sequence every facet of the genome, including the historically elusive telomeres and centromeres [31,32]. Long-read whole-genome sequencing (WGS) stands out for its capability of navigating through intricate genomic regions, facilitating the identification of structural variations like insertions, deletions, inversions, translocations, expansions, and copy number variations [33,34]. Short-read sequencing, with its typically shorter read lengths, struggles to capture these structural variations adequately [34]. The exploration of structural variation through long-read WGS may shed light on some of the unexplained heritability in ALS [35]. Until now, only one study has delved into long-read whole-genome sequencing (WGS) within the realm of ALS, specifically targeting C9orf72 repeat expansions. Utilizing the ONT MinION (Oxford Science Park, Oxford, UK), no reads covering the C9orf72 expansion were identified, while PacBio SMRT sequencing yielded an 8× coverage of the expansion [35]. Notably, there have been no reported large-scale association studies in ALS integrating long-read WGS.

## 3. Genetic Mutations and Corresponding Cellular Alterations in Neurodegenerative Disorders

In this section, we will delve into the cellular mechanisms of pathogenicity linked to genetic variations, focusing specifically on neurodegenerative disorders such as ALS, AD, and PD. Additionally, other neurodegenerative diseases are also touched on briefly in this section to provide a comprehensive overview of this expansive field.

### 3.1. Amyotrophic Lateral Sclerosis (ALS)

ALS is a fatal neurodegenerative disease that predominantly affects motor neurons in the brain, brainstem, and spinal cord [6]. The term “amyotrophy” refers to the loss of muscles and “lateral sclerosis” refers to the loss of axons and muscle in the lateral spinal cord columns. ALS causes gradual voluntary muscle weakness that spreads to neighboring body parts, usually resulting in death from respiratory failure within 2–4 years of diagnosis. In addition to motor neuron loss, the neuropathological features include intracellular cytoplasmic inclusions of eosinophilic Bunina bodies and ubiquitinated TDP-43 [6]. Furthermore, there is significant variation in disease symptoms, with frontotemporal dementia (FTD) occurring in about 15% of patients, and cognitive and behavioral changes occurring in up to 60% of patients [6].

#### 3.1.1. Genetic and Pathological Overlap between ALS and FTD

ALS has a close association with frontotemporal dementia (FTD), sharing a common molecular etiology with this condition [36]. Roughly 15% of individuals diagnosed with FTD develop motor neuron disease (MND), while conversely, up to 50% of those with MND exhibit clear signs of cognitive defects [37]. Numerous studies have revealed clinical, pathological, and genetic similarities between these conditions. Consequently, they are now viewed as two expressions of a single disease continuum, known as either the ALS-FTD spectrum or the FTD-ALS spectrum. The broader term, frontotemporal lobar degeneration, is linked with motor neuron disorders under the name FTLD-MND [36]. ALS is a degenerative MND marked by progressive muscle atrophy, paralysis, and ultimately, death typically occurring within 3–5 years of symptom onset due to respiratory failure. On the other hand, FTD manifests with alterations in social behavior and/or language skills at disease onset, stemming from neurodegeneration in the frontal and temporal lobes, culminating in death within 3–12 years of symptom onset [38].

Despite the heterogeneous clinical phenotypes observed in FTD and ALS, implying potentially divergent underlying biological mechanisms, it is widely acknowledged that these diseases share significant clinical, genetic, and neuropathological similarities [39]. The prevalent mutations implicated in disease causation are found in chromosome 9 open reading frame 72 (*C9ORF72*) and progranulin (*GRN*), both leading to TDP-43 neuropathology, as well as in microtubule-associated protein tau (*MAPT*), which leads to tau neuropathology [40]. The leading factors responsible for ALS, FTD, or their co-occurrence (ALS-FTD), in a familial context, are commonly the pathogenic hexanucleotide repeat expansions in C9ORF72 [41]. Approximately 25–40% of familial cases of ALS and FTD exhibit this mutation, while 5–7% of sporadic cases also test positive for pathogenic expansions in C9ORF72 [41]. In addition to *C9ORF72*, mutations in several other genes, such as *TARDBP*, *SQSTM1*, *VCP*, *FUS*, *TBK1*, *CHCHD10*, and *UBQLN2*, have been identified in association with both ALS and FTD [34,37]. For patients with ALS and FTD who possess these particular mutations, excluding FUS, the prevalent pathological feature is the presence of ubiquitinated protein deposits primarily consisting of TDP-43 [36]. The reason why mutations in the same genes lead to distinct clinical syndromes despite similar neuropathology remains unclear and could be attributed to variations in mutation localization affecting downstream processes, as well as potential influences from modifying genetic and/or environmental factors. However, the presence of shared genetic factors in FTD and ALS suggests the existence of common molecular mechanisms driving disease pathology, to some degree. Ongoing efforts in developing novel disease-modifying treatments for both ALS and FTD patients are targeting specific molecular subtypes.

#### 3.1.2. Epidemiology

The risk of ALS increases with aging and is highest between the ages of 60 and 79 [42]. It is unclear whether the incidence of ALS has changed in the last few decades, but it is expected to rise as the population ages. According to a meta-analysis, the standardized global incidence of ALS is only 168 per 100,000 person-years of follow-up, but this varies by region [42]. In populations with a predominance of European ancestry, like those in Europe and North America, the incidence ranges from 1.71 per 100,000 to 1.89 per 100,000 and may even be higher in population-based studies [43]. Asian populations have lower incidences, varying from 0.73 per 100,000 in south Asia to 0·94 per 100,000 in west Asia, whereas Oceania universally has the highest incidence (2.25 per 100,000) [43]. Additionally, incidence differs by sex, with a standardization male-to-female ratio of 1.35 that is influenced by the age of onset. Genetics also play a role; heritability is higher in mother–daughter pairs, whereas the most common known ALS risk gene, *C9orf72*, lowers the onset age in men versus women [43]. Thus, ALS is caused by complex interactions between age, gender, and genetics, which have implications for preclinical and clinical research.

Even though there are several known genetic risks for ALS, approximately 85% of cases do not have a single genetic cause [44]; thus, the pathophysiology of the disease remains unknown, delaying the development of effective therapies. Till now, there are only two effective drugs available: riluzole and edaravone [45]. Recently, two additional drugs were approved by the FDA that include Toferson (the first gene therapy for ALS) [46] and Relyvrio [47]. Non-pharmacological multidisciplinary care, for example, the use of early non-invasive ventilation and feeding tube insertion, can improve patient outcomes to a certain extent before significant weight loss [6]. Due to the scarcity of treatments, researchers have focused their efforts on the complex genetics of ALS and the associated pathomechanism [48].

#### 3.1.3. Genetic Causes and Risk Factor

ALS is typically classified as familial or sporadic. This straightforward division, however, ignores the disease’s complex genetic architecture, which is characterized by gene penetrance, heritability, and inheritance (monogenic, oligogenic, and polygenic). Only 10–15% of individuals have Mendelian ALS, though it has incomplete penetrance in most families [49]. In the remaining 85%, large GWASs may be able to find rare variants, i.e., mutations found in a single family that may affect disease risk and phenotypic presentation [50]. Ancestral European (i.e., European, American, Canadian, and Australian) and Asian populations are the main sources of the current knowledge of validated genes for ALS [51]. Although at least 40 genes have been linked to the disease, 4 genes, namely *C9orf72*, *SOD1*, *TARDBP* (coding for TDP-43), and *FUS*, account for roughly 48% of familial cases and 5% of sporadic cases in European populations, and significantly contribute to the pathophysiology of disease [52].

Superoxide dismutase (*SOD1*)

In 1993, the identification of the *SOD1* gene as the causative factor for the familial form of ALS was a major advancement in this field [53]. SOD1 (Cu-Zn SOD) is a 32 kda homodimeric protein, comprising 153 amino acids and containing one copper and one Zn binding site, present abundantly in the nucleus, cytosol and mitochondria [54]. Currently, more than 200 mutations are reported in this gene [55]. SOD1 plays antioxidant role in the cellular system by lowering the concentration of reactive oxygen species (ROS), and also converting ROS into oxygen and hydrogen peroxide (H_2_O_2_) [56].

Patients with a *SOD1*-mutation have clinical traits like early onset, longer disease duration, and motor symptoms that typically start in the lower limbs, with rare occurrences of cognitive disturbances [57]. The discovery of *SOD1* in ALS has significantly added to the understanding of the disease, in particular with the development of transgenic models. Mutated *SOD1* triggers ROS production and causes oxidative stress and abnormal iron metabolism, but the detailed molecular mechanism is still to be elucidated [58]. The mutated *SOD1* contains an exposed N-terminal short domain, the derlin-1-binding region (DBR), that initiates endoplasmic reticulum stress. In ALS, *SOD1* mutation induces a conformational change that may lead to motor neuron toxicity [59].

Excitotoxicity (glutamate-mediated neurotoxicity) is a possible pathogenic mechanism for ALS, which explains why the glutamate release inhibitor riluzole shows modest therapeutic efficiency in this disorder [60]. Even though this is debatable in *SOD1* ALS patients, excitotoxicity may result from a selective loss of the glutamate transporter-1 (GLT1), excessive glutamate efflux, or toxicity of glial cells, the consequences of which are disturbances in neuronal calcium homeostasis [58,61]. Mitochondrial dysfunction has also been observed in *SOD1*-mutated ALS animal models; an increase in the production of oxidative stressors such as nitric oxide, superoxide, and peroxynitrite, as well as a decrease in the ability of mitochondria to synthesize adenosine triphosphate (ATP) are mitochondrial-based mechanisms causing motor dysfunction [62]. Depolarization is a crucial phenomenon in axonal transports. Motor neurons are particularly sensitive to energy reduction because it disturbs ionic Na^+^/K^+^ pumps and quickly results in a slow depolarization with hyperexcitability due to the progressive loss of Na^+^ and K^+^ reversal potentials [63]. The depolarization is further worsened by persistent sodium channels (pNa^+^) and raises intracellular calcium that triggers apoptotic pathways [63]. Interestingly, Riluzole is a pNa^+^ blocker and this property may also play a role in its neuroprotective effects [64].

Recently, some authors have proposed the ‘prion-like propagation’ of mutant SOD1 misfolding and motor neuron degeneration along corticospinal pathways [65]. The concept driving this hypothesis is that a mutant or wild-type misfolded protein spreads along anatomical pathways and transmits its abnormal misfolding properties to native proteins, causing toxic aggregation. These properties have been reported in vitro for *SOD1*, *TARDBP*, *FUS*, and *C9ORF72*, but in vivo only for *SOD1* so far [66].

Moreover, the genome-wide data showed that the disease progression and severity are correlated with *SOD1* point mutations, like the p.A4V mutation is associated with severe disease symptoms and p.H46R shows slow progression [67,68]. In homozygous cases, a fast disease progression is reported in the p.L126S mutation [69], which is relatively slow in heterozygous cases [70]. Recently, another *SOD1* (p.L144S) variation in an Iranian family was reported as severe in a homozygous condition [71]. The relatively common mutation p.N86S shows low penetrance and phenotypic diversity even within families [72].

Chromosome 9 open reading frame 72 (*C9ORF72*)

The *C9ORF72* gene is present at the 9p21 locus of chromosome 9. In 2011, a massive GGGGCC (G4C2) hexanucleotide repeat expansion mutation (HREM) within intron 1 of *C9ORF72* was recognized as a pathogenic mutation in ALS [73]. Nearly 30 G4C2 HREMs have been reported in healthy individuals, whereas more than 70 repeats are speculated in ALS patients [73]. Presymptomatic carriers of the *C9ORF72* expansion mutation have earlier brain atrophy (especially focal atrophy of the left supramarginal gyrus) and cognitive alterations than healthy controls of similar ages [74]. A few theories have been put forth to explain how *C9ORF72* contributes to the emergence of *C9ORF72*-related disorders. The first theory says that a decrease in the *C9ORF72* protein level inhibits endosomal trafficking and perturbs endocytosis, which results in impaired autophagy; however, *C9ORF72* knockout mice do not exhibit motor neurodegeneration [75]. Another theory postulates that the massive G4C2 HREMs could be neurotoxic by forming length-dependent RNA foci that would enclose RNA binding proteins and shut down the RNA processing system [76]. According to a third hypothesis, the C9/ALS pathogenicity is mediated by dipeptide repeats (DPRs) which are derived from repeat-associated non-AUG translation of G4C2 (or G2C4) RNA in five different DPRs called ‘polyGA’, ‘polyGP’, ‘polyGR’, ‘polyPA’, and ‘polyPR’ [77]. The aggregated DPRs have been reported in the motor neuron of an ALS patient, but their role in disease progression is not clear [77]. *C9ORF72* expansion mutations result in susceptibility to Ca^2+^ permeable-amino-3-hydroxy-5-methyl-4-isoxazolepropionic acid (AMPA) receptor-mediated excitotoxicity and cause motor neuron degeneration [78]. *C9ORF72* repeat expansion results in mutant protein and haploinsufficiency from the wild-type allele. Therefore, in *C9ORF72* ALS patients, haploinsufficiency for *C9ORF72* activity results in neurodegeneration, which is caused by at least two mechanisms: accumulation of glutamate receptors (which results in excitotoxicity) and impaired clearance of neurotoxic DRPs derived from the repeat expansion. Additionally, *C9ORF72* expansion RNA transcripts aggregate into toxic RNA foci, blocking RNA-binding proteins and altering RNA metabolism. The abnormal translation of *C9ORF72* transcript expansions produces toxic dipeptide repeats, e.g., poly proline-arginine repeats (poly-PRs) and poly glycine-arginine repeats (poly-GRs) [79].


*TARDBP*


In 2008, *TARDBP* (encoding TDP-43) was recognized as a causative gene of ALS [80]. TDP-43 aggregates in the cytoplasm of motor neurons were reported in approximately 90% of sporadic and familial (with *C9ORF27* mutation) cases, and hence considered as a key pathological feature in ALS. The cytoplasmic aggregates of TDP-43 are characterized by abnormal phosphorylation, truncation, and mislocalization. The accumulation of abnormal TDP-43 or loss of function in physiological TDP-43 causes neurodegeneration in the ALS disease. TDP-43 propagates in a prion-like fashion along the motor pathway [81]. Moreover, even if TDP-43 is common in both ALS and FTD, in ALS, the inclusions are circumferential (curved) in contrast to FTD where the inclusions are rounded.

Familial ALS with *TARDBP* mutations is more common in the limbs and has a wider age range of onset. Among these mutations, the p.G376D mutation has a notably fast disease progression, taking less than 1.5 years from onset to death [82]. The p. G298S is another mutation considered as short-lived [57], whereas the p.A315T mutation has a longer disease course of 8–10 years [57].

*TARDBP*-coded TDP-43 is a ribonucleoprotein involved in exon splicing, gene transcription, mRNA stability, mRNA biosynthesis, and the formation of nuclear bodies [83]. The C-terminal region of TDP-43, which is rich in Gln/Asn-sequence like prions, is capable of binding directly to several ribonucleoproteins mediating protein–protein interactions and splicing repression [84]. TDP-43 loses the ability to mediate splicing repression when its C-terminal region is lacking [84]. TDP-43 accumulation results in RNA instability that triggers apoptosis by disrupting the pathways for energy production and protein synthesis [85]. TDP-43 cytoplasmic inclusions are nearly the most common feature, reported in about 97% cases of ALS [48]. It is a nuclear DNA/RNA binding protein that mislocalized to the cytoplasm and underwent significant post-translational modification or truncation in patients with ALS. TDP-43 mislocalization impairs the RNA splicing of stathmin-2, a protein required for microtubule stability, lowering stathmin-2 concentration, and impairing axonal growth and motor neuron function [86].


*FUS*


*FUS* was identified as a causative gene for ALS in 2009 [87]. In the early onset of ALS, the frequency of *FUS* mutation is high because of de novo mutation [88]. The *FUS* mutations typically manifest in patients in their 30s or 40s, with upper extremity or cervical onset and a fast-progressing disease course of 2 years [89]. In contrast, p.Q519E and p.S513P are reported mutations showing older age of onset and slow disease progression [55].

TDP-43 neuronal inclusions are widely accepted pathological hallmarks of both ALS and FTD; however, it has been demonstrated that 10% of FTD patients also have inclusions in their neuronal and glial cells that are immunoreactive for FUS but lack TDP-43 inclusions [90]. The cytoplasmic FUS inclusions comprise RNAs and proteins (from suspended translation units) and promote cell survival under stressed environments by redistributing translational resources. FUS are also involved in synaptic plasticity and dendritic integrity of neurons [91]. In the presence of mutated *FUS*, the function of stress granules (FUS inclusions) is compromised and results in motor neuron dysfunction [92].

#### 3.1.4. Additional Risk Loci from Genome-Wide Association Studies

In recent years, new genes for ALS have been identified, including *TBK1*, *NEK1*, *CCNF*, *C21ORF2* (also known as *CFAP410*), *ANXA11*, *TIA1*, *KIF5A*, *GLT8D1*, *LGALSL*, and *DNAJC7*, which have highlighted significant recurrent pathways and opened new research directions [48]. Importantly, the genes linked to ALS have a different degree of pathogenicity and susceptibility risk. Highly penetrant mutations (e.g., in *TARDBP*, *SOD1*, and *FUS*) usually cause disease, whereas some variants associated with ALS (e.g., *ANG*, *ATXN2*, and *DCTN*) do not necessarily cause the disease, rather increase the risk of its development [48]. Moreover, even causative mutations are not completely penetrant, and interactions with the environment alter the risk of disease development. A recently published GWAS, including 29,612 patients with ALS and 122,656 controls, identified 15 risk loci (Table 2) [93]. Among the newly discovered genes in ALS, *TIA1* participates in RNA metabolism; *TBK1*, *CCNF*, and *NEK1* are involved in proteostasis or autophagy; and *ANXA11*, *C21orf2*, and *KIF5A* are involved in cytoskeletal or trafficking defects [94]. The mechanism of neurodegeneration mediated by *DNAJC7*, *GLT8D1*, and *LGALSL* is not clear. It is speculated that *DNAJC7*, a heat shock protein co-chaperone, may be involved in proteostasis or autophagy, and *GLT8D1*, a glycosyltransferase, may interfere in ganglioside biosynthesis and O-linked β-N-acetylglucosamine modification [48]. The cellular role of galectin-related protein (encoded by *LGALSL*) is unknown; however, galectins are galactoside-binding proteins.

Thus, the identification of novel genes for ALS may open unexplored research fields and pathological mechanisms [48]. Despite significant progress in ALS research, the pathophysiology is still only partially understood. However, as we obtain more insights into the genetic architecture of ALS, the molecular mechanisms by which various mutations converge on recurrently dysregulated nervous system pathways are being discovered. Impaired RNA metabolism, trafficking defects, autophagy, mitochondrial dysfunction, and compromised DNA repair are the most common pathological pathways reported in ALS patients [79]. Among the most prevalent ALS genes, mutant *C9ORF72*, *TARDBP*, and *FUS* all impair RNA metabolism; *C9ORF72* repeat expansions, *TARDBP*, and *SOD1* cause defects in protein homoeostasis and autophagy.

### 3.2. Alzheimer’s Disease (AD)

Alois Alzheimer, on 3 November 1906, in the 37th Meeting of the South-West German Psychiatrists, reported an unusual case study involving a “peculiar severe disease process of the cerebral cortex” and this study was later published in 1907 [109]. Alzheimer described a 51-year-old woman with paranoia, progressive sleep and memory disturbance, aggression, and confusion in his seminal paper. Less than 5 years after she passed away, a neuropathologic examination of her brain revealed the plaques and tangles that are now considered as the main pathological feature of the disease [110]. With shared underlying neuropathologic changes, AD is now understood to be a heterogeneous and polygenic group of both hereditary and sporadic neurodegenerative disorders. Even though there are many different variants of AD, most cases still have a clinical presentation like that of the original case, albeit with a later age of onset. A clinical diagnosis is frequently made based on initial memory dysfunction that later stretches to affect multiple cognitive domains. A neuropathologic diagnosis of AD requires the presence of both amyloid beta (Aβ) plaques outside neuronal cell, and neurofibrillary tangles (NFTs) inside the neurons, which can now be easily identified using immunohistochemical stains such as Aβ and tau, respectively [111]. Alzheimer’s initial discoveries laid the groundwork for the current clinical and neuropathologic diagnostic criteria for AD; however, over time, our understanding of the underlying genetics has led to changes in the conceptualization and characterization of this disease.

The Aβ plaques and NFTs had previously been reported in the brains of elderly subjects both with and without dementia, and some researchers considered them a part of the aging process [112]. What set Alzheimer’s case apart was that it occurred in a relatively young individual; therefore, AD was defined as a separate disease from what was observed in older people. In the second half of the 20th century, however, AD became redefined as a clinically heterogeneous disease, united by a common underlying set of pathologic changes in the brain that could affect more commonly older but also younger adults [113].

Based on an age cutoff of 65 years, AD is commonly classified as either early onset (EOAD) or late onset (LOAD) [114]. LOAD cases have some genetic risk factors but often occur sporadically and are by far the most common. On the contrary, EOAD is responsible for only 5–10% of all AD cases [115]. In general, two types of inheritance patterns have been noticed in EOAD: mendelian (mEOAD) patterns and nonmendelian (nmEOAD) patterns. mEOAD forms are fully penetrant, with an autosomal dominant inheritance pattern and frequent mutations in *APP*, *PSEN1*, and *PSEN2*. nmEOAD, on the other hand, is sporadic or has irregular inheritance patterns (i.e., inheritance patterns that are not obviously autosomal dominant, or with highly variable age at onset, including LOAD). The genetic etiology of nmEOAD is unknown, but it is widely assumed to be polygenic and multifactorial in nature [114].

#### 3.2.1. Epidemiology

AD is the most prevalent type of dementia, accounting for 50–75% of all cases [114]. According to current estimates, 5.4 million Americans have AD; by the middle of the next century, that number is expected to quadruple, largely due to population aging [116]. It is estimated that by 2050, a new case of AD will be diagnosed every 33 s, resulting in nearly a million new cases each year [116]. After age 65, the incidence of AD rises exponentially with age and doubles every five years. Currently, the disease affects one in nine people over the age of 65 and one in three people over the age of 85 [116]. While deaths from heart disease, prostate cancer, and stroke have decreased over the past ten years, the number of deaths from AD have increased, making it the sixth leading cause of mortality in the United States. By the year 2050, it is anticipated that the total cost of health-care, long-term care, and hospice services for those with dementia will have increased from the current USD 290 billion to more than USD 1.1 trillion [116].

While the field still has challenges in defining the etiology, diagnosis, and treatment of AD, significant progress has been made in our comprehension of the role of genetics in LOAD as well as in the sporadic and familial EOAD forms. It is a consensus that a precision medicine strategy will be necessary to treat this complicated and multifactorial disease, which requires a highly sophisticated understanding of its genetic frameworks.

#### 3.2.2. Genetic Causes and Risk Factors

In 1991, Goate et al. discovered a missense mutation on exon 17 of the *APP* gene that changed amino acid 717 from valine to isoleucine (p. Val717Ile) [117]. The mutation is known as the London mutation because it was first identified in an English family. Following this initial finding, Chartier-Harlin et al. and Murrell et al. discovered different mutations (V717G and V717F, respectively) in the same amino acid in additional families with AD [118,119]. Despite being in ethnically Romanian kindred, the mutation discovered by Murrell et al. became known as the Indiana mutation. In 1993, Mullan et al. identified two large Swedish families with EOAD who had a double mutation (K670N/M671L) in the N terminus of *APP* called the Swedish mutation [120]. Since then, numerous additional *APP* mutations have been found, directing research towards the *APP* processing pathways [121].

The known *APP* mutations were unable to account for the presence of EOAD in several families. The first non-*APP* mutations in the *presenilin 1* (*PSEN1*) gene were discovered in multiple ethnic families in 1995 [122]. These included *PSEN1* C410Y (Ashkenazi Jewish), H163R (American and French Canadian), M146L (Italian), L286V (German), and A246E (Anglo-Saxon–Celt). Shortly after this finding, Campion et al. found six novel *PSEN1* mutations in eight additional families, all of which were in highly conserved regions of the gene [123]. Since then, numerous additional *PSEN1* mutations have been identified [124]. A third case of the *PSEN1* variant, Y389H, has been identified with EOAD in a Korean patient [125]. The identification of *presenilin 2* (*PSEN2*), a gene homologous to *PSEN1* on chromosome 1, was the next major genetic discovery in AD. Rogaev et al. [126] discovered two different mutations that were associated with familial AD namely M239V in Italian kindred and N141I in a Volga German pedigree, while Levy-Lahad et al. [127] also discovered the N141I mutation in Volga German kindred. With the advancement in sequencing technology, Finckh et al. sequenced individuals with EOAD and discovered additional mutations in *PSEN1* (F105L) and *PSEN2* (T122P and M239I), respectively [128]. Furthermore, known mutations in these three genes, *APP*, *PSEN1*, and *PSEN2*, account for only about 1% of autosomal dominantly inherited AD [129], but their identification has resulted in a significant advancement in understanding of the pathophysiology of AD. According to estimates, between 55–75% of LOAD cases are heritable. In contrast to EOAD, no causal genes for LOAD have been found; instead, several risk genes have been identified [114].


*APOE*


The gene *APOE* on chromosome 19 encodes apolipoprotein E, and unlike all other mammals, humans have three prevalent alleles (ε2, ε3, and ε4). Pericak-Vance et al., using an affected-member-pedigree linkage analysis, discovered that the patients’ genomic markers were linked to chromosome 19 (HAS 19) rather than to the previously identified chromosome 21 locus [130]. Simultaneously, Namba et al. found that apoE was localized to NFTs and Aβ deposits in the brain of AD patients [131]. Strittmatter et al. used an in vitro assay in 1993 to demonstrate that apoE binds to Aβ with high avidity and that *APOE-4* was more prevalent in LOAD patients than in unrelated, age-matched controls [132]. Following this, Corder et al. evaluated 42 LOAD families and discovered a gene dosage effect, resulting in approximately a 3-fold higher risk in *APOE ε4/ε4* genotype (homozygous) compared to those with a single *APOE ε4* allele [133]. Even though the frequencies of *APOE* genotypes vary across ethnic groups, the ε4 allele has been repeatedly linked to an increased risk of AD. Interestingly, in a large Colombian EOAD cohort, a carrier of the *PSEN1* (E280A) mutation also had two copies of the rare *APOE3* Christchurch mutation (R136S), and she did not experience cognitive deficits until her 70s, nearly three decades after the usual age of onset [134]. This discovery has sparked interest in the potential protective benefits of this mutation.


*SORL1*


Following the discovery of the gene dosage-dependent risk of *APOE ε4* with LOAD, researchers investigated other genes in the endocytic and cellular recycling pathways. Scherzer et al. discovered six genes that exhibited differential expression in AD patients using a DNA microarray screen [135]. One of those genes was *SORL1*, which encodes a neuronal apoE receptor. Six SNPs were found by Rogaeva et al. to be significantly associated with AD in two different regions of the *SORL1* gene using previously identified SNPs in these pathways [136]. Intronic variants in *SORL1* have been linked to familial and sporadic AD, according to Lee et al. [137], and the TGen data set and an urban multiethnic community have confirmed these findings [135]. These intronic variants are thought to be in regulatory sequences and may affect SORL1′s physiological role in APP processing.


*MAPT*


Multiple NDDs have been linked to NFTs which are encoded by microtubule-associated protein tau (*MAPT*). Several missense mutations, insertions, deletions, and splice-site mutations were found in the *MAPT* gene, which was initially studied in relation to FTD [138]. Roks et al. performed an association study using the A169 polymorphism in exon 9 and the (CA) n-repeat polymorphism in intron 9 and discovered no mutations causally associated with EOAD [139]. Baker et al. sequenced the *MAPT* gene to look for an association between *MAPT* and progressive supranuclear palsy (PSP) [140]. They discovered a series of SNPs that were completely out of sync with one another, revealing two extended haplotypes (H1 and H2) that covered the entire *MAPT* gene. Further research showed that Caucasian patients with tauopathies have an overrepresentation of the H1 haplotype [141]. However, subsequent research on the *MAPT* haplotype and AD has produced mixed results, with some studies finding a link between the H1 haplotype and AD and others failing to support these findings [142]. Since the H2 haplotype is almost exclusively found in Caucasian populations, Sun and Jia evaluated only the H1 haplotype in a Chinese Han population and discovered an SNP within the promoter of *MAPT* [143]. Comparing patients with sporadic AD from the Chinese Han ethnic group to healthy controls revealed that the 347C allele was overrepresented and when tested in cell lines, the SNP (347C/C versus 347 G/G) significantly increased transcriptional activity, upregulating gene expression [143].


*TREM2*


Triggering receptor expressed on myeloid cells 2 (*TREM2*) is a receptor of the innate immune system expressed on microglia, macrophages, dendritic cells, and encoded by *TREM2* [144]. *TREM2* mutations were discovered in the early 2000s to cause polycystic lipomembranous osteodysplasia with sclerosing leukoencephalopathy (PLOSL), also known as Nasu–Hakola disease [145]. Presenile dementia with neurological abnormalities is among the prominent symptoms of PLOSL. However, *TREM2* was not thought to be a potential candidate gene for dementia until *TREM2* mutations were discovered in a Lebanese family with early-onset dementia but no PLOSL [146]. Following the discovery of *TREM2*’s potential role in early onset dementia, Guerreiro et al. studied whole exome/genome sequencing of 281 AD patients and 504 healthy controls. They observed a disproportionate number of variants in exon 2 of *TREM2* in AD patients compared to healthy controls, and these variants increased the AD risk in a heterozygous state [147]. So far, 46 genetic variants in *TREM2* have been investigated in relation to LOAD. Within each population, these variants cause a roughly 2–4-fold increase in the risk of developing LOAD [148]. Additionally, *TREM2* variants have been linked to ALS, PD, and FTD indicating that altered *TREM2* function may indirectly increase the risk of neurodegeneration, possibly through dysfunctional microglia [149].


*ABCA7*


ABCA7 is a membrane-bound protein associated with transporting lipids across the cell membrane through the utilization of energy derived from ATP. It plays pivotal roles in three key cellular processes: regulating cholesterol metabolism, managing phospholipids, and facilitating phagocytosis [150,151]. ABCA7 facilitates the formation of lipid and exchangeable apolipoproteins, including apolipoprotein E (apoE), into high-density lipoprotein (HDL) particles, which are then discharged into the extracellular environment. This process, known as cell lipid efflux, ultimately results in the removal of lipids from the cells [152]. ABCA7 is involved in AD pathogenesis, contributing to the clearance of Aβ and the transport of the amyloid-β protein precursor (AβPP) [151]. Disruptions in the *ABCA7* gene due to mutations can impair one or more functions of the protein, potentially leading to the development of the neuropathology associated with AD. However, mutations in *ABCA7* do not have uniform effects, and the precise alterations in *ABCA7* function resulting from these mutations associated with AD risk remain elusive. Understanding these variations is particularly crucial due to the significant impact of ABCA7-related AD risk among African American/Black adults [153]. In mouse models of AD, the absence of *ABCA7* leads to elevated levels of Aβ in the brain, either by enhancing Aβ production or impairing its clearance [153]. Both mouse Abca7 knockouts and human carriers of the *ABCA7* AD-risk allele, who do not have AD, generally demonstrate minor behavioral and cognitive alterations [154].

#### 3.2.3. Additional Risk Loci from Genome-Wide Association Studies (GWASs)

GWASs have been a more recent method for determining genetic susceptibility to AD. A total of 695 genes have been found to affect the risk of LOAD through 1395 GWASs and 320 meta-analyses [155]. According to the GWASs, the top 10 genes that are most strongly linked to the risk of LOAD are *APOE*, *BIN1*, *CLU*, *ABCA7*, *CR1*, *PICALM*, *MS4A6A*, *CD33*, *MS4AE*, and *CD2AP* [155]. Several of these studies combined multiple GWAS data sets with the idea that a common disease might present with a common variant, necessitating a large number of samples. A more recent, two-stage GWAS involving 111,326 clinically diagnosed/“proxy” AD cases and 677,663 controls was recently published. They discovered 75 risk loci (Table 3), 42 of which were novel at the time of the study [156].

To validate *CLU*, *CR1*, and *PICALM* as risk loci for LOAD, Jun et al. examined the SNPs within each gene in various populations (Caucasian, African American, Arab, and Caribbean Hispanic), as well as in a combined data set [224]. They stated that the *CLU*, *CR1*, and *PICALM* loci in this population are susceptibility loci for AD. Additionally, the authors examined these SNPs collectively with *APOE* genotypes and discovered that the association with *PICALM* was only seen in patients who had the *APOE 4* allele, whereas the *CLU* was only present in patients lacking the *APOE 4* allele [224]. Kunkle et al. confirmed many previously identified genome-wide significant loci in 94,437 LOAD patients (*CR1*, *BIN1*, *INPP5D*, *HLA-DRB1*, *TREM2*, *CD2AP*, *NYAP1*, *EPHA1*, *PTK2B*, *CLU*, *SPI1*, *MS4A2*, *PICALM*, *SORL1*, FERMT2, *SLC24A4*, *ABCA7*, *APOE*, and *CASS4*) [225]. These genes may have a pleiotropic connection with traits associated with AD. The authors further compared the genetic makeup of LOAD to 792 other human illnesses, traits, and behaviors to further explore this theory [225]. They confirmed the findings of Marioni et al. that the genetic architecture of LOAD is more strongly associated with a maternal family history of AD than with a paternal family history of AD [226]. Kunkle et al. performed a pathway analysis for common and rare variants found in LOAD patients to better understand how all these genetic risk loci may contribute to AD. The majority of genes belonged to the gene ontology pathway for immune response activation; the assembly of protein–lipid complexes was the most significant common variant pathway, while tau protein binding was the most significant rare variant pathway [225].

Overall, NFTs and amyloid plaque deposition are considered as the major pathway in AD progression. It is noteworthy that newly discovered genetic risk factors are frequently first assessed in relation to established pathways; therefore, much research is needed to understand the underlying mechanisms in AD progression which may also avail novel therapeutic options.

### 3.3. Parkinson’s Disease (PD)

PD is a complicated and progressive neurodegenerative disorder first described by James Parkinson in his 1817 publication, “Essay on the Shaking Palsy” [227]. In that essay, Dr. Parkinson expressed hope by writing, “there appears to be sufficient reason for hoping that some remedial process may ere long be discovered, by which, at least, the progress of the disease may be stopped” [227]. Until now, no definitive neuroprotective therapy for PD has been developed after more than 200 years.

However, significant advances in our understanding of the molecular basis of neurodegeneration in PD have been made in recent years, bringing us closer to developing effective disease-modifying treatments. Pathologically, PD is characterized by the loss of dopaminergic neurons and deposition of α-synuclein aggregates (Lewy bodies) in the substantia nigra pars compacta (SN) located in the midbrain [228]. Additional brain areas and nondopaminergic neurons are also included in PD pathology. However, recent research has indicated that the loss of dopaminergic terminals in the basal ganglia, rather than neurons in the SN, is critical for the onset of motor symptoms [229]. Moreover, symptomatically, the disease is characterized by resting tremors, cogwheel rigidity, bradykinesia, autonomic dysfunction, and cognitive decline. Additionally, anosmia, constipation, depression, and rapid eye movement (REM) sleep behavior disorder could be noticed long before the motor dysfunction in PD patients [230].

#### 3.3.1. Epidemiology

According to health-care utilization estimates, the annual incidence of PD ranges from 5/100,000 to more than 35/100,000 new cases [231]. Age is a significant risk factor, and the prevalence of PD increases with age. The prevalence of PD is anticipated to rise sharply as the world’s population ages, doubling in the following two decades [232]. In a meta-analysis of four populations in North America, it was found that prevalence increased from less than 1% of men and women aged 45–54 to 4% of men and 2% of women aged 85 or more [229]. Along with this increase, the societal and economic burden of PD will also rise unless more effective treatment options, or prevention methods, are discovered.

Most PD cases are likely to have a multifactorial etiology caused by the interaction of environmental and genetic factors. PD risk may be increased by head trauma and toxic chemical exposure, while it may be decreased by certain lifestyle choices. Although 5–10% cases of PD can be linked directly to identifiable mutations in particular genes, most PD patients lack these mutations [233]. The most prevalent PD-linked genetic mutations also have incomplete penetrance, suggesting the involvement of additional genetic or environmental factors. Simon et al. (2020) compared the prevalence rate in monozygotic and dizygotic twins and found that the heritability of PD is only 30%, implying that most of the PD risk is related to environmental and behavioral factors [233].

#### 3.3.2. Genetic Causes and Risk Factors

PD genetics provides an important window into the disease’s mechanism by highlighting specific molecular pathways and their cellular effects. The intricate genetics of PD are outlined in this section, with an emphasis on the genes that have been most extensively studied.


*SNCA*


*SNCA* (PARK1/4) is a five-exon gene present on chromosome 4 (4q22.1) and is one of three paralogs that comprise the synuclein family. Synucleins are presynaptic proteins and include α-synucleins, β-synucleins, and γ-synucleins encoded by *SNCA*, *SNCB*, and *SNCG,* respectively [234]. *SNCA*-encoded α-synuclein is a 14-kDa protein and expressed ubiquitously in the brain. It has been shown to promote the long-term maintenance of the presynaptic compartment, aid in the formation of SNARE complexes, and control the release of dopaminergic vesicles [234]. β-Synuclein shares 78% structural homology with α-synuclein and is co-expressed with α-synuclein throughout the brain, including the presynaptic region. Meanwhile, γ-synuclein exhibits 67% homology and is found in the peripheral nervous system, specific central neuronal subtypes, and in breast, colon, and pancreatic cancers [234]. Even though only α-synuclein is thought to be involved in the pathophysiology of PD, patients with DLB (dementia with Lewy bodies) have been found to have lower levels of the *SNCB* transcript, and in vitro studies indicate that β-synuclein may antagonize α-synuclein-induced aggregation and toxicity [235]. Point mutations in *SNCA* were discovered to be the first cause of autosomal dominant PD [236], leading to the discovery that α-synuclein is the most abundant protein component of Lewy bodies (LBs) [237]. Other reported pathogenic SNCA variants, in addition to p.A53T, are p.A30P, p.E46K, and p.G51D. These missense mutations are all found in the membrane-binding region of α-synuclein [238].

Additionally, there is intrafamilial heterogeneity, which affects how people with *SNCA* mutations present clinically. Generally, all patients have an early onset of the disease (as young as 19 years old), with the majority of cases developing symptoms when patients are 40 to 60 years old [239]. All patients present with the traditional motor symptoms of PD. However, patients with p.A53T, p.E46K, or p.G51D mutations also have early cognitive impairment. Pathologically, these patients exhibit widespread LB accumulation, SN degeneration, cortical thinning (p.A53T), and involvement of the motor cortex, among other features (p.G51D) [239].

A disease-causing triplication of the wild-type *SNCA* locus was described shortly after the discovery of these point mutations in a family with autosomal dominant parkinsonian motor signs, early onset of symptoms (mean age 36 years), and prominent cognitive defects [240]. Complete loss of the SN and severe cortical and hippocampal atrophy were present in the four patients from this family who underwent neuropathological examination [240]. Later, more than 19 families have been reported with autosomal dominant disease caused by duplications in wild-type *SNCA* [241]. The identification of these families provides several important insights into the pathophysiology of PD. It first suggests that increases in wild-type α-synuclein levels are enough to trigger PD pathogenesis. Second, the disease symptoms are linked to gene dosage, with *SNCA* triplication patients experiencing symptoms sooner than duplication patients [242]. Thirdly, this implies that PD risk may be raised by less penetrant *SNCA* variants that result in small changes in α-synuclein levels. This latter hypothesis is supported by a study published in 2016, which discovered a variant in the *SNCA* enhancer linked to an elevated risk of PD, that reduces the binding of transcriptional repressors and increases *SNCA* expression [243]. 


*LRRK2*


Autosomal dominant PD is also caused by variations in *LRRK2* (*PARK8*) [244]. The age of disease onset differs between *LRRK2* pathogenic variants and families [242]. Leucine-rich repeat kinase 2 (LRRK2) is a 253-kDa protein, encoded by the *LRRK2* gene and expressed in the brain, lungs, kidneys, and immune system [239]. It has both GTPase and kinase activity, with the GTPase activity being carried out by a Ras of complex proteins (ROC)/C-terminal of Ras (COR) and the kinase activity being carried out by the kinase domain [245]. The remaining domains of this protein participate in interacting with other proteins. Up to 1% of all cases of sporadic PD are caused by mutations in *LRRK2*, the second most frequent genetic risk factor. *LRRK2* variants account for 29% of familial PD in Ashkenazi Jews [246] and 70% of PD in North Africans [247]. All the highly penetrant *LRRK2* variants are thought to increase the kinase activity directly or indirectly. Later, a clinical report documented an increased LRRK2 kinase activity in sporadic PD patient brains and provided additional evidence to support this hypothesis [248]. This, coupled with the absence of any PD-like phenotypes in *Lrrk2^−/−^* mice, has led to the hypothesis that elevated kinase activity is what causes *LRRK2* to have a pathogenic effect [249]. *LRRK2*-PD patients have a clinical phenotype like sporadic PD, but the pathology varies [250]. Given the complexity of how *LRRK2* mutations affect the biology of the protein, it is perhaps not surprising that different mutations cause different pathological phenotypes or that they cause incomplete penetrance.


*VPS35*


*VPS35* (PARK17) encodes vacuolar protein sorting 35 (VPS35), a heteropentameric protein complex that is responsible for retrograde protein sorting from the endosome to the cell membrane and trans-Golgi network, and vice versa. The c.1858G>A (p.D620N) variant, which is found primarily in Caucasian populations, accounts for an estimated 0.1–1% of patients with familial PD and causes autosomal dominant PD with high but incomplete penetrance [251]. Patients with *VPS35* p.D620N present clinical symptoms similar to those seen in idiopathic PD, with an average age of onset ranging from 48.3 to 53 years old, depending on the family [252]. It remains to be seen whether this variant has a dominantly negative effect, a toxic gain-of-function, or another mechanism. Patients with *VPS35* variants may or may not have α-synuclein pathology. This is because the only data available are based on a single autopsy with only a few pathological investigations [251].


*GBA*


*GBA* is the gene that encodes β-glucocerebrosidase (GCase), a lysosomal enzyme that degrades glucosylceramide (GluCer) into glucose and ceramide. Homozygous *GBA* mutations cause Gaucher’s disease, a recessive lysosomal storage disorder (GD) [253]. This disease primarily affects the liver, spleen, and bones, but it can also affect the central nervous system. GD patients are classified based on brain involvement, with type 1 patients having no neurological symptoms and types 2 and 3 experiencing neurological symptoms [253].

However, depending on the variant and population studied, heterozygous *GBA* mutation carriers have a 5–8-fold increased risk of PD [253]. The risk of PD or DLB is increased by *GBA* variants in heterozygosity that result in GD in homozygosity [254]. *GBA* variants are the most prevalent genetic risk factor for PD. Two *GBA* variants, namely E365K and T408M, raise the risk of but do not result in GD [255,256]. Clinically, people with PD having mild *GBA* mutations exhibit motor symptoms that are like those of idiopathic PD and respond to levodopa quickly. The clinical symptoms, however, are related to the relevant variant, with some patients showing early disease onset like sleep disorders, dementia, and a higher frequency of psychiatric symptoms such as depression and anxiety [257]. The brains of these patients show typical neuropathological features of PD, such as nigral degeneration and LB deposition throughout the brain [257]. 

#### 3.3.3. Additional Risk Loci from Genome-Wide Association Studies

In 2009, the first GWAS loci for PD were identified using information from about 5000 patients and 9000 healthy controls [258]. To date, the largest GWAS of PD was carried out including the study of 7.8 M SNPs in 37.7 K cases, 18.6 K UK Biobank (Cheshire, UK) proxy-cases (having a first degree relative with PD), and 1.4 M healthy controls. Consequently, 90 distinct independent risk signals (Table 4) were identified in 78 genomic regions across the entire genome, including 38 novel independent risk signals in 37 loci [259].

Numerous GWAS loci have been found to be close to the so-called monogenic PD gene (*SNCA*, *LRRK2*, *GBA*, and *VPS35*); these regions are referred to as pleomorphic risk loci [326]. PD can be caused by rare coding variants at these genes, but more common variants—often non-coding and with a smaller effect size—can also raise the risk of developing the condition. Additionally, some loci are near genes that contribute to the development of additional conditions, including *MAPT*, *GRN*, *NEK1* linked to FTD and ALS, and *NOD1* linked to Crohn’s disease and Blau syndrome [326]. According to estimates, the heritable portion of PD caused by common genetic variability is approximately 22% [327], and the GWAS loci discovered to date only account for a small portion of this. So, there are still a ton of undiscovered risk variants.

Overall, understanding the genetic factors influencing PD risk, onset, and progression is critical for developing treatments that can slow or stop disease progression. Many genes and GWAS loci that contribute to the development of PD have been identified so far. The search for genetic risk factors must continue, and a concerted, cooperative effort must be made to comprehend the implications of these findings on the molecular and biological levels.

Based upon the discussion so far, it is clear that genetic factors play a significant role in increasing the risk of neurodegenerative diseases and influencing the expression of disease characteristics; however, only about 10% cases of NDDs are considered to have familial form of disease, and among these cases only a fraction can be attributed to known, rare, highly penetrant genetic variants. Similarly, while genome-wide association studies (GWASs) have made notable progress in identifying common GWAS-SNPs in cohort with neurodegenerative disease, despite the advancement, these variants account for only a small portion of heritable risk [328,329]. Even after combining the effect of both Mendelian large-effect rare mutations and common disease-associated SNPs, a considerable portion of heritability across NDDs remains unexplained [330,331], suggesting its multifactorial nature and the involvement of environmental factors.

As a result, identifying particular risk factors, relevant biomarkers, prospective new therapeutic targets and agents, and even definitive diagnoses remain difficult. Pathological brain features during neurodegeneration demonstrate a significant overlap across distinct forms of neurodegenerative disorders. There is currently no diagnostic test that can clearly establish the existence, absence, or categorization of a neurodegenerative disorder. The underlying mechanism is another unsolved subject in the majority of neurodegenerative disorders. Most can be identified by intracellular protein deposits; however, it is unclear whether this is a key mechanism or a result of another disrupted cell function. There are numerous proposed pathways of neurodegeneration, including primary impacts of protein homeostasis, disrupted protein degradation, mitochondrial dysfunction, and so on (Figure 3). It is critical to better understand the disease pathophysiology to enhance early diagnosis and the development of disease-modifying medicines.

#### 3.3.4. Other Neurodegenerative Disorders

Huntington’s disease (HD) manifests as a rare monogenic neurodegenerative condition marked by motor, cognitive, and psychiatric deficits. These symptoms typically arise in patients aged 30–50 years and death occurs approximately 10–15 years after the onset of clinical symptoms [332]. Huntington’s disease (HD) results from the expansion of a CAG repeat region in exon 1 of the huntingtin (HTT) gene, situated on chromosome 4, surpassing a pathogenic threshold of at least 37 CAGs. Inheritance of 40 or more CAGs within this region is linked with 100% disease penetrance [332]. While Huntington’s disease (HD) is primarily attributed to inheriting at least one mutant HTT allele, there is growing recognition of additional genetic factors that influence the age-of-onset and severity of HD. These factors are currently under investigation as potential targets for therapeutic interventions [333]. The expansion of CAG repeats results in the production of mutant HTT (mHTT) proteins containing elongated N-terminal polyQ tracts. These expanded tracts directly induce misfolding of the mHTT protein [334]. Misfolded mHTT proteins aggregate into amyloid structure, forming intranuclear inclusion bodies, which are a distinctive diagnostic hallmark observed in the brain of individuals with HD [335]. The misfolding and aggregation of HTT disturb various cellular processes, supported by significant evidence suggesting that oligomers and/or insoluble fibrils of mHTT play a direct role in causing neurodegeneration in HD [334]. HTT knockout is embryonic lethal, while conditional knockout induces progressive neurodegenerative phenotypes in adult mice. This highlights the indispensable role of HTT expression in the normal development of the CNS [336].

Prion diseases represent a distinctive and uncommon set of neurodegenerative disorders found naturally in both humans and various animal species. Creutzfeldt-Jakob disease (CJD) in humans, scrapie in sheep and goats, bovine spongiform encephalopathy (BSE) in cattle, and chronic wasting disease (CWD) in cervids are just a few examples of well-known forms of prion disease [337]. Both human and animal prion diseases share a common pathology, featuring spongiform degeneration in the grey matter regions of the brain, reactive proliferation of glial populations, neuronal loss, and the accumulation of a misfolded and disease-associated form of the prion protein known as PrP^Sc^ in the CNS [338]. Human prion diseases, like other neurodegenerative disorders, primarily occur as sporadic or genetic conditions. Notably, sporadic CJD is the most prevalent form, constituting roughly 85% of all human prion diseases [337]. The revelation that prion diseases are linked to the conversion of the normal host-encoded cellular prion protein (PrPC) into a misfolded form (PrPSc) through post-translational modification, regardless of nucleic acid involvement, is recognized as the protein-only hypothesis [338]. A polymorphism at codon 129 (c129) within the prion protein gene (PRNP) significantly impacts both susceptibility to and the clinical characteristics of human prion diseases [338].

Multiple sclerosis (MS) is a chronic inflammatory and neurodegenerative disease of the CNS, mediated by the immune system [339]. MS stands as a primary contributor to disability among young adults and is estimated to impact approximately 2.8 million individuals globally [340]. GWASs have uncovered over 230 risk alleles associated with MS, the majority of which are genes involved in regulating the immune system [341]. The initial event triggering symptomatic pathology involves the infiltration of peripheral immune cells previously sensitized to elements of the myelin sheath. A key characteristic of MS is the formation of focal inflammatory and demyelinating lesions, primarily occurring in white matter regions of the brain, optic nerve, and spinal cord, although lesions in intracortical and deep gray matter are also observed [339]. T and B lymphocytes, along with macrophages, predominantly migrate from the periphery into the CNS parenchyma, resulting in the emergence of perivascular demyelination, subependymal/pial demyelination, and neuroaxonal degeneration. Inflammatory demyelination and compromised healing ultimately lead to axonal transection, resulting in permanent neurodegeneration and clinical disability [339]. MS is characterized by three primary disease courses: relapsing–remitting (RRMS), primary progressive (PPMS), and secondary progressive (SPMS). In each clinical course, patients may encounter fluctuating disease activity, manifested by the emergence of new or worsening neurological dysfunction [342].

## 4. Gene Therapy for Neurodegenerative Diseases

In concept, gene therapy is a simple and clear-cut process. It entails treating a disease through the introduction of a transgene that either replaces or repairs a malfunctioning gene [343]. In practical application, the process is significantly more intricate, requiring optimization of various factors like selecting the appropriate vector, optimizing the delivery method, and carefully choosing the transgene are pivotal in practice. The dynamics between the host’s immune system and the vector or transgene can introduce further complexities to the overall therapeutic approach. In neurodegenerative diseases, the intricacies of the target tissue add an additional layer of complexity.

In the domain of gene therapy vectors, there are two primary classifications: viral and non-viral [344]. Viral vectors utilize the inherent infective properties of viruses, undergoing genetic modifications to eliminate replicative genes, making them suitable for clinical applications [344]. In addressing neurodegenerative diseases, the prominent viral vectors include adeno-associated viruses (AAVs) and lentiviruses [345]. Both AAVs and lentiviral vectors can effectively infect dividing and non-dividing cells. However, a key distinction lies in the fact that lentiviruses integrate into the host genome, whereas AAVs do not [345]. Integration provides enduring and stable expression but simultaneously introduces the potential risk of integrational mutagenesis. Even though AAVs do not integrate, they can still facilitate persistent gene expression in nondividing cells [345]. Non-viral vectors typically involve either naked plasmid DNA or complexes with cationic lipids or polymers. These vectors exert a localized impact and demand a higher therapeutic dosage compared to viral vectors [346]. Generally, non-viral delivery results in only temporary gene expression, often inadequate for effectively addressing chronic neurodegenerative conditions [346].

Choosing the delivery route is crucial, especially for the central nervous system (CNS). Remote delivery achieved through intravenous injection, as a non-invasive approach, is advantageous. However, the blood–brain barrier (BBB) stands as a substantial barricade impeding the entry of most vectors into the CNS [347]. Therefore, the significant discovery that AAV9 can penetrate the blood–brain barrier (BBB) is notable [348]. Remote delivery poses certain drawbacks, including an elevated risk of off-target effects and the necessity for a larger dose to attain a therapeutic level in the target tissue. Conversely, direct delivery to the CNS mitigates off-target effects and lowers the necessary dosage of the gene therapy vector [347]. In the central nervous system (CNS), this can be achieved through intraparenchymal injection (directly into the brain or spinal cord) or by injecting into the cerebrospinal fluid (CSF), either through intracerebroventricular (ICV) or intrathecal routes [347].

The resurgence of gene therapy brings with it the enticing promise of not only targeting the fundamental causes of disease but also providing enduring corrections [349]. Severe combined immune deficiency (SCID) was the first clinical success with gene therapy. SCID is the most severe human inborn errors of immunity, with absent T and B lymphocyte function, making the infant susceptible to life-threatening infections with high mortality in the absence of treatment [350]. Additionally, Casgevy is the first FDA-approved gene therapy to utilize a type of novel genome-editing technology (CRISPR-Cas9) for the treatment of sickle cell disease (SCD) in patients 12 years and older [351]. In neurodegenerative diseases, spinal muscular atrophy (SMA) was the first to be cured with the exogenous introduction of SMN1 gene to children under two years of age [352].

Embracing various forms of genome manipulation, this approach is particularly attractive in the domain of neurodegenerative diseases, where traditional pharmacological interventions have faced consistent challenges [345]. In contrast to organs where achieving therapeutic concentrations through repeated doses is more feasible, most agents administered peripherally encounter difficulties in efficiently crossing the blood–brain barrier when targeting the brain. Hence, the notion of a singular, enduring intervention, often termed “one and done”, becomes particularly attractive when addressing diseases impacting the central nervous system (CNS) [353]. In the following discussion, we will briefly explore relevant technologies for therapeutics targeting neurodegenerative diseases.

## 5. Gene Expression

### 5.1. The Exogenous Introduction of Genes into the CNS

The most direct approach in gene therapy involves the exogenous introduction of genes into the CNS. These introduced genes have the potential to rectify the loss of gene function caused by pathological mutations, such as the instance of the SMN1 gene in spinal muscular atrophy [352].

### 5.2. DNA Editing

The clinical adoption of DNA-editing tools is underway, with their ability to modify gene expression or address pathogenic mutations [354]. In broad terms, two crucial elements come into play: DNA-binding domains designed to identify specific genomic sequences and nucleases that create double-stranded breaks (DSBs). DSB repair is facilitated by nonhomologous end-joining (NHEJ), an endogenous mechanism prone to errors, resulting in the insertion and deletion of sequences (INDELs) within the reading frame [354]. Typically, this leads to frameshift mutations and premature termination codons (PTCs), ultimately causing the targeted gene to be knocked out. In an alternative scenario, when an external template is available, intrinsic homology-dependent repair (HDR) mechanisms come into play to introduce desired sequences or point mutations into the host genome [354]. Three fundamental DNA-editing nucleases include zinc-finger nucleases (ZFNs), transcription activator-like effector nucleases (TALENs), and CRISPR-associated nucleases [355]. The structure of ZFNs involves two essential segments: a DNA-binding domain composed of three–six zinc fingers, each with the capability to recognize three DNA base-pairs in the host genome, and a DNA-cleaving domain originating from the endonuclease Fok1 [355]. Given that Fok1 functions as a dimer, the design involves two ZFNs binding to opposite strands of the intended genomic DNA. This configuration enables the Fok1 domains to dimerize and initiate DNA cleavage [355]. In the case of the transcription activator-like effector nucleases (TALENs), they incorporate a sequence of transcription activator-like effectors (TALEs) along with the Fok1 DNA-cleavage domain [356]. The TALE polypeptide is made up of 33 or 34 amino acids, and within this structure, residues 12 and 13 are responsible for recognizing a specific DNA base [356].

In the synthetic CRISPR system, a custom-designed guide RNA (gRNA) is employed to direct a Cas9 nuclease to specific locations on the host DNA (Figure 4). Another prerequisite involves the presence of protospacer adjacent motif (PAM) sites, which are scattered throughout the genome [357]. For the frequently used Cas9, streptococcus pyogenes Cas9 (SpCas9), the corresponding guide RNA (gRNA) is comprised of a CRISPR RNA (crRNA) recognizing the genomic target sequence and a trans-activating CRISPR RNA (tracrRNA) that facilitates the recruitment of Cas9 [357]. Following the introduction of DSBs by Cas9 and subsequent repair through NHEJ, INDELs commonly lead to a shift in the reading frame, introducing a PTC [358]. Typically, mRNAs containing these mutated PTCs undergo degradation via nonsense-mediated decay (NMD), an intrinsic surveillance mechanism, resulting in the depletion of the corresponding protein. Theoretically, CRISPR-guided HDR offers the potential to introduce specific point mutations or insertions into the native genome using donor templates (the flow-chart is shown in Figure 5). However, the frequency of HDR-mediated repair is generally lower than that of NHEJ in most cells, restricting the scope of recombination [353]. Additionally, HDR naturally transpires during mitotic recombination, limiting its application primarily to postmitotic neurons. Despite these challenges, ongoing strategies are being developed to enhance HDR efficiency in the brain [359].

### 5.3. CRISPR-Mediated Base Editing and Prime Editing

The prevalent genetic disorders in human pathologies, encompassing inherited neurodegenerative diseases, are attributed to single-base pair point mutations [360]. DNA base editors are hybrid proteins formed by the fusion of a dCas9 or nCas9 with a deaminase protein, enabling the deamination and modification of cytidine or adenine base pairs (Figure 6). Cytosine base editors facilitate the transformation of C:G to T:A, while adenine base editors alter A:T to G:C [361,362]. The effective use of base editing encounters limitations due to the necessity for appropriate PAM sequences and the occurrence of undesired nucleotide substitutions in the vicinity of the target site, arising from the broad activity range of deaminases. Recent Cas9 variations have been created exhibiting increased PAM compatibility and cytidine deaminase variants with more restricted activity windows [183]. Overcoming the obstacles associated with the large size of the base editors for AAV packaging in in vivo delivery, a recent study employed dual AAVs. These dual AAVs transported split base editors, which were subsequently reassembled in situ through trans-splicing inteins [363].

Off-target effects, particularly undesired RNA editing, have been documented in the context of base editing [364]. Nevertheless, more recent versions of base editors have been reported to demonstrate reduced off-target activity [365]. The concept of prime editing (Figure 7) broadens the scope of base editing, offering expanded genomic targeting, more possibilities for genetic alterations, and enhanced precision. The prime-editing guide RNA (pegRNA) is designed to encompass both the typical DNA-targeting sequence and the desired editing sequence, allowing for the substitution of targeted genomic DNA nucleotides [366]. The associated nCas9 is coupled with a reverse transcriptase, initiating the reverse transcription of pegRNA (Figure 8). This process transfers the encoded genetic information from the pegRNA, encompassing insertions, deletions, and base conversions, to the targeted genome [366].

## 6. Genetic Therapy for AD

One of the initial clinical trials in gene therapy for Alzheimer’s disease involved introducing the gene encoding nerve growth factor (NGF) to the cholinergic nucleus basalis of Meynert. This was accomplished through direct, bilateral injections of AAV2 into the brain [367]. NGF, as an intrinsic neurotrophic factor, governs the growth and viability of cholinergic neurons by functionally activating the TRKA receptor [368]. Studies conducted in preclinical animal models, including non-human primates (NHPs), demonstrated the favorable impact of administering external NGF, thereby establishing a rationale for human trials [369]. Despite the safety and sustained expression of AAV2–NGF observed in Phase 1/2 clinical trials for mild-to-moderate AD, there was no obvious improvement in cognitive function [370]. Furthermore, analysis of autopsy brains from this study revealed that the administered AAV2–NGF did not reach the intended cholinergic neurons in the nucleus basalis of Meynert [371]. Consequently, the effectiveness of this therapeutic approach remains uncertain.

### 6.1. Targeting MAPT

Utilizing antisense oligonucleotides (ASOs) against the MAPT gene, which encodes tau, researchers have effectively reduced MAPT mRNA and tau protein levels in animal models. These ASOs demonstrate the ability to mitigate tau phosphorylation and aggregation, prevent neuronal death, and extend survival in transgenic mice expressing a human mutant (P301S) tau. Moreover, they contribute to a reduction in CNS tau levels in NHPs [372]. This strategy is presently being tested in a clinical trial and reported more than 50% reduction in tau synthesis in the CNS of mild AD patients (NCT03186989) [372]. In an alternative approach, ASO-mediated exon skipping can effectively lower MAPT mRNA and tau protein levels in both cellular and in vivo settings [373]. Emerging evidence indicates that tau is involved in regulating presynaptic function [374] and the acute knockdown of tau in adult mice led to learning and memory deficits [375].

### 6.2. Targeting APOE

Human susceptibility to Alzheimer’s disease is linked to three apolipoprotein E (APOE) polymorphic alleles—E2, E3, and E4. APOE4 is recognized as the primary risk factor for sporadic Alzheimer’s disease, while APOE2 is associated with a protective effect [376]. Research with animal models implies an association between APOE4 and the progression of both amyloid beta and tau pathology [377]. A gene therapy approach involves increasing the levels of protective APOE2 in the brain. Indeed, expressing APOE2 through viral vectors attenuates Aβ pathology in a mouse model that relies on amyloid [378]. A Phase 1 trial involving the delivery of AAV–APOE2, administered into the cerebrospinal fluid through intracisternal injections, is anticipated to commence shortly (NCT03634007). Human APOE4 and APOE3 exhibit a distinction in only one amino acid residue at position 112 [379]. While current technology faces limitations in producing widespread single-nucleotide changes in the brain, research has demonstrated that converting APOE4 to APOE3 through gene editing in induced pluripotent stem cells (iPSCs) or cerebral organoids can alleviate AD-associated phenotypes [380]. This approach holds promise as a potential option for gene therapy in the future.

### 6.3. Targeting APP

The APP gene plays a pivotal role in both sporadic and familial AD, making the silencing or modulation of APP an attractive prospect for gene therapy. Employing CRISPR–Cas9 technology, a recent investigation selectively deactivated mutant familial APP Swedish alleles in both cellular and in vivo settings, without impacting the corresponding wild-type alleles [381]. In an alternate study, antisense oligonucleotides (ASOs) were employed to omit the penultimate exon (exon 17) of APP [382]. The γ-secretase cleavage site and a substantial portion of the transmembrane domain of APP are encoded by exon 17. Deleting this exon results in the removal of the membrane-anchoring segment of APP, leading to a decrease in Aβ secretion in both cellular and in vivo settings [382]. Nonetheless, the elimination of the membrane-anchoring segment of APP is anticipated to result in the loss of its physiological functions. In a recent investigation, a CRISPR–Cas9 DNA-excision approach based on NHEJ was employed to introduce PTCs in the last exon (exon 18) of APP. This resulted in the truncation of the last approximately 36 amino acids and the removal of a pentapeptide YENPTY endocytic motif that initiates the amyloidogenic pathway [383]. Regulatory mechanisms in transcription render the last exon impervious to NMD. Consequently, the presence of PTCs in this region does not lead to mRNA degradation; instead, it causes protein truncations. This method carries the additional benefit of preserving the integrity of the amino terminus and transmembrane domains of APP, which are thought to contribute to axonal and synaptic physiology. As a result, there is no observable impact on physiological functions [383].

## 7. Gene Therapy for PD

### 7.1. Modulating Neuronal Signaling

Enhancing dopaminergic signaling in Parkinson’s disease (PD) has been achieved through gene therapies utilizing adeno-associated viruses (AAVs). The regulation of the dopamine synthesis pathway involves three key enzymes—GTP cyclohydrolase 1 (GCH1), tyrosine hydroxylase (TH), and aromatic amino acid DOPA decarboxylase (AADC). Clinical trials involving AAV2–*AADC* have shown safe, sustained expression over a period of up to 4 years, and a modest amelioration of symptoms [384]. A more precise administration of AAV2–*AADC* in non-human primates (NHPs) through real-time MRI guidance demonstrated safety and good tolerability, as evidenced by the trial results (NCT03065192) [385]. Furthermore, a gene therapy utilizing lentivirus, known as ProSavin, was designed to deliver all three rate-limiting enzymes (TH, AADC, and GCH1). The therapy demonstrated positive tolerability during Phase 1/2 trials, and subsequent examinations indicated moderate improvements in motor function [386]. The decline in dopaminergic tonus within the striatum in PD results in an overactive glutamatergic subthalamic nucleus (STN). Preclinical studies demonstrated that enhancing GABA activation via the subthalamic overexpression of glutamate acid decarboxylase (GAD), an enzyme involved in GABA synthesis, effectively alleviated PD-like symptoms [387]. In Phase 1/2 clinical trials, the administration of AAV2–*GAD* into the STN demonstrated both good tolerability and improvement in symptoms associated with PD that persisted for a year [388].

### 7.2. Targeting Disease Genes—SNCA, GBA, and LRRK2

In rodent models of PD, the use of short hairpin RNA (shRNA) or antisense oligonucleotides (ASOs) to knockdown α-synuclein has proven effective in preventing neurodegeneration [389,390]. Additionally, CRISPR-mediated gene silencing of α-synuclein boosted the cell viability in dopaminergic neurons derived from human-induced pluripotent stem cells (iPSCs) sourced from a patient with PD [391]. However, unresolved concerns have been noted in some in vivo studies regarding physiological phenotypes following the knockdown of α-synuclein [392].

Administering AAV–*GBA1* directly into the brain resulted in a reduction in α-synuclein levels and pathology in rodent models [393]. When GBA1 was diffusely delivered to the brain through intravenous injections of AAV-PHP.B–*GBA1* in an A53T α-synuclein mouse model, it not only alleviated α-synuclein pathology but also contributed to substantial behavioral recovery and an extension of lifespan [394]. A recent Phase 1/2 clinical trial (NCT04127578) is underway to treat individuals with PD. The approach involves the intracisternal injection of AAV9–*GBA1* directly into the cerebrospinal fluid.

The expression of *LRRK2* in the lungs, kidneys, and spleen implies that a general inhibition of *LRRK2* may give rise to pathological modifications in these specific tissues [395]. Hence, in this context, gene therapy proves beneficial by selectively inhibiting *LRRK2* in the brain. The successful attenuation of mRNA and protein levels through the intracerebral injection of *LRRK2* ASOs was achieved without noticeable phenotypic alterations in the kidneys or lungs [396]. Intrathecal injections of BIIB094, an ASO targeting *LRRK2* developed by Ionis Pharmaceuticals, is currently in a Phase 1 clinical trial for individuals with PD (NCT03976349).

## 8. Gene Therapy for ALS

### Targeting SOD1

Administering ASO–*SOD1* into the cerebrospinal fluid (CSF) effectively diminished SOD1 levels and significantly prolonged survival in a transgenic rat model expressing mutated human *SOD1* [397]. Furthermore, the intrathecal administration of ASO–*SOD1* has been well-tolerated among patients with ALS exhibiting *SOD1* mutations [398]. A follow-up Phase 1/2 study demonstrated that the administration of ASO–*SOD1* resulted in a reduction in SOD1 protein levels in CSF [399] and a Phase 3 trial is currently underway (Tofersen). Additionally, RNA interference (RNAi) and CRISPR technology have been employed to downregulate mutant *SOD1;* a one-time intrathecal or intravenous injection of AAV-*SOD1*-sgRNA results in reduction in SOD1 levels, enhanced motor function, and significantly extended the lifespan of mice having mutant *SOD1* [400]. Recently, a proof-of-concept investigation demonstrated the feasibility of intrathecally delivering AAVs expressing microRNAs against *SOD1* in two patients with *SOD1*-related ALS. Postmortem analysis in one patient revealed the effective suppression of *SOD1* in the spinal cord, affirming the potential success of this approach [401]. The intracerebroventricular injection of ASO–*C9orf72* resulted in a reduction in *C9orf72* mRNA foci and cognitive improvement in a transgenic mouse model with 450 G4C2 repeats in the *C9orf72* gene [402]. An ongoing Phase 1 clinical trial is investigating the intrathecal administration of ASO–*C9orf72* in patients diagnosed with *C9orf72*-linked ALS.

## 9. Conclusions and Future Perspectives

The discussion so far has concluded that the neurodegenerative diseases discussed in this review, including AD, PD, and ALS, share several epidemiologic and genetic features. First, they all have an etiologic dichotomy, with more common late-onset forms and less common familial forms. It is possible (and likely) that a significant proportion of cases that were previously thought to be sporadic and nonfamilial will ultimately be found to result from disease-causing mutations or genetic risk factors (like APOE-ε4 in AD). Second, in some instances, identical mutations and polymorphisms have been linked to and associated with a variety of clinically and neuropathologically distinct disease entities. GWASs have played a significant role in this effort because they have enabled the discovery of novel genetic associations that are not based on prior knowledge. Unfortunately, GWAS findings, so far, have explained only a small proportion of the heritability of complex diseases, making genetic risk prediction tests for these diseases currently unfeasible. Overcoming the constraints of GWAS, NGS platforms like WGS and WES have revealed a plethora of rare variants that are pivotal in understanding complex neurological diseases and Mendelian neurological conditions.

In the past, tackling the pervasive pathology of neurodegenerative diseases through gene therapy was deemed challenging. However, breakthroughs in vector technologies now enable the broad delivery of genes into the CNS. Coupled with modern genome-manipulation tools, gene-based therapies hold the potential to revolutionize the clinical management of both inherited and sporadic neurodegenerative diseases—conditions marked by their devastating impact and a current lack of effective disease-modifying treatments. Nevertheless, numerous challenges persist, and the pivotal transition into a new era requires systematic advancements in gene-delivery vector development, comprehensive safety assessments of contemporary gene manipulation tools, and transparent collaboration among various stakeholders.

## Figures and Tables

**Figure 1 ijms-25-02320-f001:**
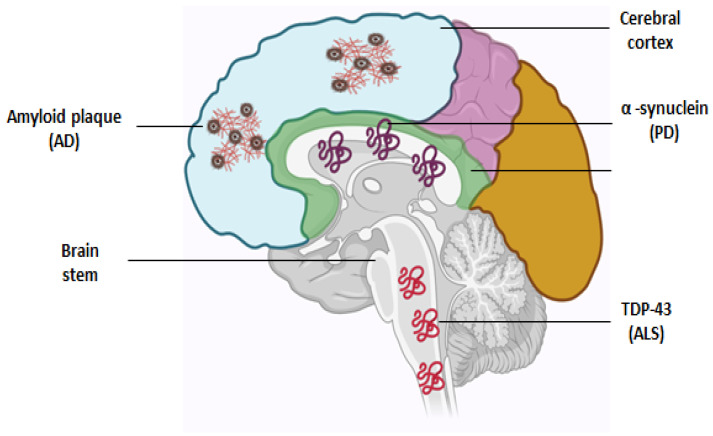
Abnormal protein deposition in affected brain regions during AD, PD, and ALS. In AD, cerebral cortex and hippocampus is predominantly affected and identified by amyloid plaque deposition, and a-synuclein deposition in basal ganglia is a hallmark of PD. In ALS, the brain stem and spinal cord are affected with the accumulation of TDP-43 aggregates.

**Figure 2 ijms-25-02320-f002:**
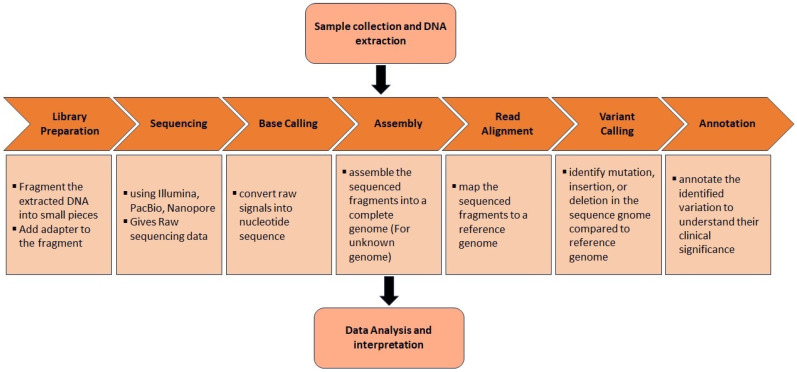
A simplified flowchart depicting the streamlined pathways involved in next generation genome sequencing.

**Figure 3 ijms-25-02320-f003:**
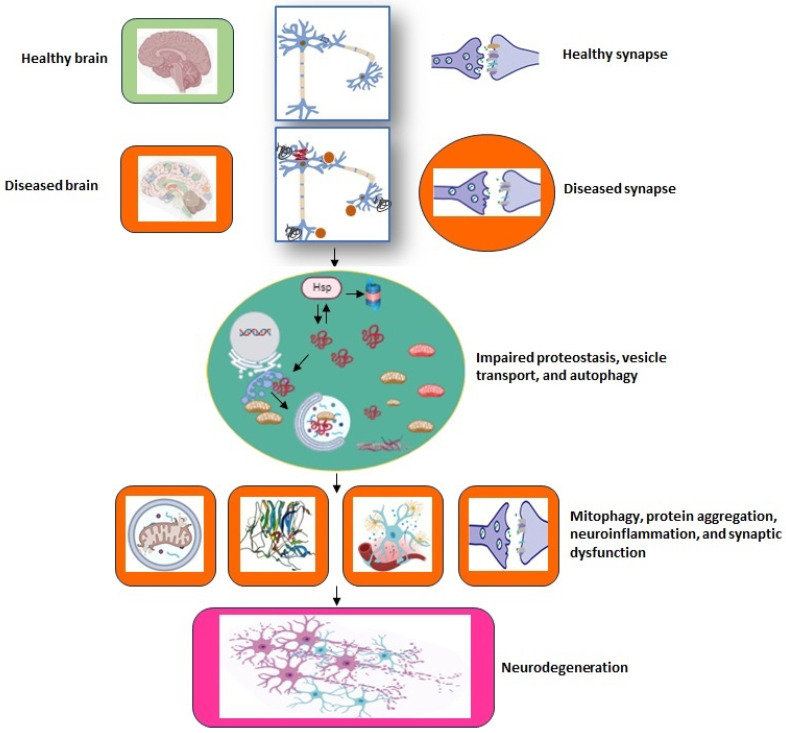
This pictorial illustration aims to provide a detailed overview of protein aggregation, the intricate molecular mechanism involved, and the resultant damage to neurons in common neurodegenerative conditions.

**Figure 4 ijms-25-02320-f004:**
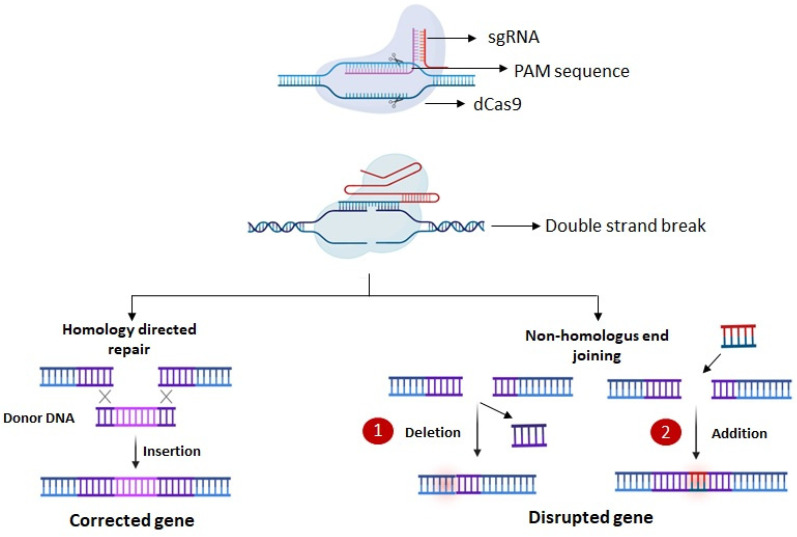
This illustration provides an overview of CRISPR-Cas9-mediated gene editing. The Cas9 enzyme is triggered by attaching to guide RNA (gRNA) and then binding to the corresponding genomic sequence just before the 3-nucleotide PAM sequence. After binding, Cas9 induces a double-strand break, and the DNA is repaired using either the NHEJ or HDR pathway, leading to an edited gene sequence.

**Figure 5 ijms-25-02320-f005:**
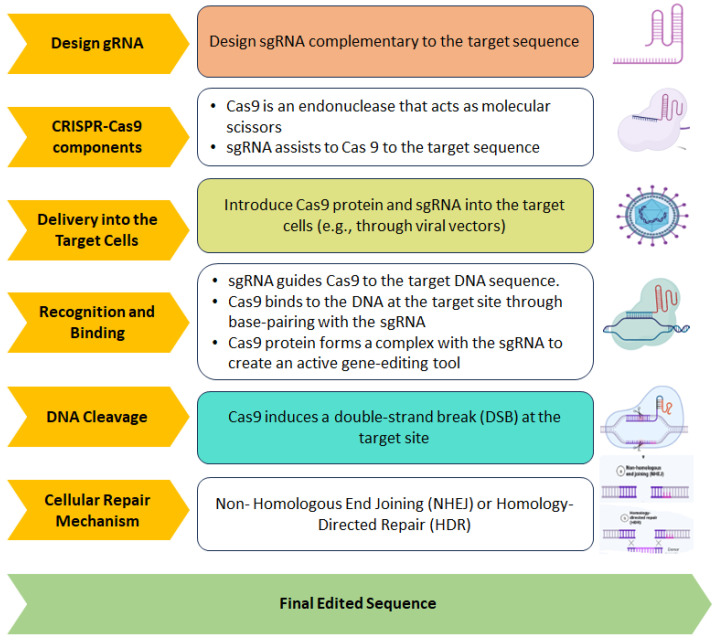
This visual illustration delineates the critical steps involved in the CRISPR-Cas9 gene editing process including the design of guide RNA, formation of the CRISPR-Cas9 complex, target DNA recognition, and the introduction of modifications.

**Figure 6 ijms-25-02320-f006:**
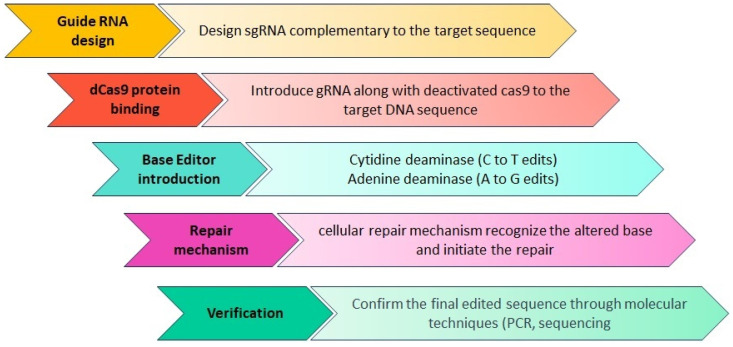
The flow chart details the sequential steps in the CRISPR-mediated base editing process, encompassing the design of guide RNA, the formation of the CRISPR base editing complex, targeted base pair modification, and the final edited genetic sequence.

**Figure 7 ijms-25-02320-f007:**
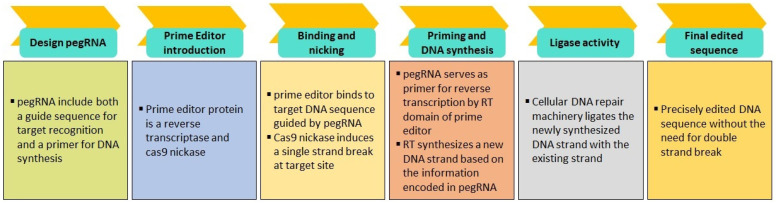
This illustrative guide delineates the critical stages in the CRISPR-mediated prime editing process including the design of prime editing guide RNA (pegRNA), formation of the CRISPR prime editing complex (reverse transcriptase and Cas9 nickase), precise editing of the target sequence and repair, and the ultimate generation of the edited genetic information.

**Figure 8 ijms-25-02320-f008:**
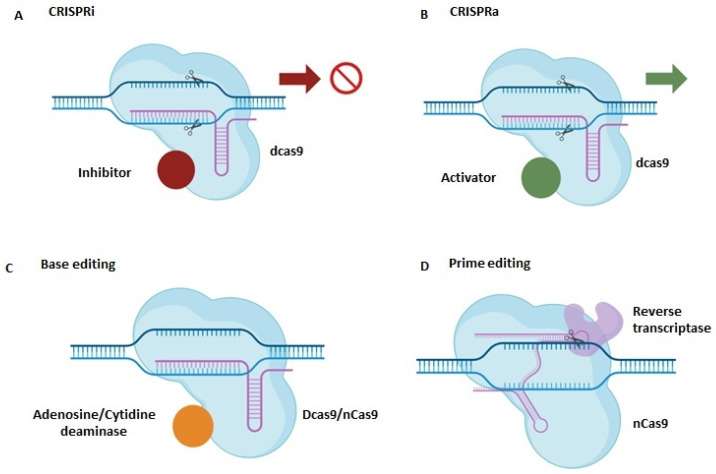
CRISPR-mediated transcriptional regulation by adding nuclease-deactivated cas9 (dcas9) to inhibit (**A**) or activate (**B**) transcription of targeted gene. CRISPR-mediated base editing involves a dCas9 or nickase Cas9 (nCas9) to bind with adenosine or cytosine (**C**) deaminase that changes AT to GC or CG to TA. The prime editing (**D**) involves nCas9 fused with RT that transcribes a part of pegRNA at target site.

**Table 1 ijms-25-02320-t001:** Comparative analysis of genetic techniques, their advantages, and limitations.

S.No.	Techniques	Methods	Benefits	Disadvantages
1	Sanger Sequencing	Traditional method involving the sequencing of DNA fragments using chain-termination dideoxy nucleotides.	▪ High accuracy in base calling ▪ Well-established and widely used sequencing method	▪ Limited throughput, suitable for sequencing suitable for sequencing small regions or individual genes ▪ Relatively higher cost per base compared to NGS
2	GWAS (Genome-wide association studies)	Analyze the genetic variation across the entire genome to identify the link between specific genetic variants and a particular trait or disease.	▪ High-throughput screening of genetic variation ▪ Can detect common variants associated with disease risk	▪ Limited ability to detect rare variants with small effect sizes. ▪ Lack of functional information about identified variants
3	WGS (Whole Genome Sequencing)	Involves sequencing the entire genome to identify both coding and non-coding variants associated with disease.	▪ Comprehensive coverage of entire genome including regulatory region ▪ Identifies both rare and common variants	▪ Higher cost and computational resources required. ▪ Challenges in interpreting non-coding variants and their functional consequences
4	WES (Whole Exome Sequencing)	Focus on sequencing the protein coding region of genome to identify disease-associated variants.	▪ Identifies rare coding variants with potentially large effect sizes ▪ Provides information on functional consequences of variants	▪ Limited coverage of non-coding regions, where regulatory variants may reside. ▪ High cost
5	NGS (Next-Generation Sequencing)	Utilizes high throughput sequencing technologies to sequence DNA or RNA molecule in parallel.	▪ Enables rapid sequencing of large amount of DNA or RNA ▪ Offers higher sensitivity and resolution compared to traditional sequencing methods	▪ Requires sophisticated bioinformatics tools and computational resources. ▪ Higher cost compared to traditional sequencing methods
6	LRS (Long Read Sequencing)	Employs sequencing platforms that generate reads spanning hundreds to thousands of base pairs, providing more contiguous sequence information.	▪ Enables sequencing of longer DNA fragments, allowing for better detection of structural variants ▪ Facilitates assembly of complex genomic region and repetitive sequences	▪ Generally lower throughput compared to short read sequencing platforms ▪ Higher errors rates in reads compared to short read sequencing

**Table 2 ijms-25-02320-t002:** Risk loci identified in cross-ancestry genome-wide association study (GWAS), the associated function and disease mechanism in amyotrophic lateral sclerosis.

S.No.	Gene (ALS)	Function	Disease Mechanism
1	*C9orf72*	Regulates vesicular transport and autophagy	C9ORF72 haploinsufficiency (loss of function) Sense and antisense RNA (GGGGCC)_n_ the function of RNA binding protein (gain of function)
2	*UNC13A*	Facilitates Neurotransmission	Impaired synaptic transmission [95]
3	*SOD1*	Antioxidant role	Oxidative stress, mitochondrial dysfunction, and excitotoxicity
4	*SCFD1*	Regulates ER to Golgi anterograde vesicular transport	Protein misfolding and aggregation [96]
5	*MOBP-RPSA*	Neurons myelination	Demyelination of neurons [97]
6	*HLA*	Antigen presentation and immune response	Inflammation due to suppressed immune response [98]
7	*KIF5A*	Engaged in anterograde transport of cargos along the microtubule rails in neurons	Impaired axonal transport, synaptic transmission, and motor neuronal toxicity [99]
8	*CFAP410*	Cytoskeletal organization and ciliary function	Decreased stability and increased degradation of mutant protein causes dysfunction of primary cilium [100]
9	*GPX3-TNIP*	Antioxidant	Oxidative stress, mitochondrial dysfunction, and excitotoxicity [101]
10	*SLC9A8*	Na/H exchanger	Excitotoxicity and axonal degeneration [102]
11	*TBK1*	Requires in cargo recruitment during autophagy	Neuroinflammation and autophagy [103]
12	*ERGIC1*	Maintains ER-Golgi structure	Disintegration of ER and mitophagy [104]
13	*NEK1*	A protein kinase that regulates cell cycle, DNA damage repair, apoptosis, and ciliary function	Induces DNA damage [105]
14	*COG3*	Regulating Golgi processes, protein trafficking, and glycosylation in neurons	Protein trafficking by Golgi fragmentation [106]
15	*PTPRN2*	Involved in vesicle-mediated secretory process in hippocampus [107]	Not clear. Probably motor neuron dysfunction [108]

**Table 3 ijms-25-02320-t003:** Risk loci identified in cross-ancestry genome-wide association study (GWAS), the associated function and disease mechanism in Alzheimer’s disease.

S.No.	Gene (AD)	Function	Disease Mechanism
1	*SORT1*	Directs trafficking of APP into recycling pathways	Low level of SORT1 in AD causes increased Aβ deposition [157]
2	*CR1*	Immune complement cascade	Regulates Aβ metabolism [158]
3	*ADAM17*	Alpha-secretase imparts a role in APP processing	Causes increased APP production [159]
4	*PRKD3*	Cell proliferation	Causes neuroinflammation [160]
5	*NCK2*	Axon growth and synapse formation and Epinephrin-mediated axon guidance	Disturbs motor axon trajectory selection [161]
6	*WDR12*	Ribosome biogenesis and cell proliferation	Possibly causing neuroinflammation [162]
7	*BIN1*	Endocytosis and intracellular trafficking	Endosome defect [163]
8	*INPP5D*	Immune signaling	Inflammasome activation in microglia [164]
9	*MME*	Cleaves and degrades beta-amyloid	Increased Aβ deposition and axonal neuropathy [165]
10	*IDUA*	Lysosomal protein acts in degradation of misfolded protein	Lysosomal dysfunction and increased proteinopathy [166]
11	*RHOH*	Regulation of actin cytoskeleton, and dendrites formation	Synaptic loss and spinal dysfunction [167]
12	*CLNK*	Immunomodulatory function	Disturbed immune signaling and neuroinflammation [168]
13	*ANKH*	Regulating inflammation	NF-κB-mediated neuroinflammation [162]
14	*COX7C*	Mitochondrial bioenergetics	Mitochondrial respiratory defects [169]
15	*TNIP1*	Inhibition of the TNF-α signaling pathway and NF-κB activation/translocation	Microglial activation and inflammation [170]
16	*RASGEF1C*	Associated with immune function	Neuroinflammation [171]
17	*HS3ST5*	Cellular uptake and distribution of molecules like growth factors and morphogens	Promotes tau fibrillation into NFTs [172]
18	*HLA-DQA1*	Dendritic cells, macrophages and B cells and involved in adaptive immune responses	Stimulates adaptive immune signaling in AD and also activates PKC and TLR signaling [173]
19	*UNC5CL*	Involved in mediating axon growth, neuronal migration in neuronal development, regulation of cell apoptosis.	Contributes to AD pathogenesis by activating DAPK1 which in turn causes aberrant tau, Aβ and neuronal apoptosis/autophagy
20	*TREM2*	Regulates microglia proliferation, survival, migration, and phagocytosis.	Downregulation induces neuroinflammation [174]
21	*TREML2*	Regulates microglial proliferation	Immune-related neuroinflammatory and increased tau deposition [175]
22	*CD2AP*	Early endosome formation and protein trafficking	Regulates Aβ generation by a neuron-specific polarization of Aβ in dendritic early endosomes [176]
23	*UMAD1*	Involved in endosome-ubiquitin homeostasis [177]	Possibly defects in protein degradation cascade and increased deposit of Aβ and tau
24	*ICA1*	ICA1 regulates AMPA receptor trafficking [178]	Possibly disturb synaptic signaling
25	TMEM106B	Brain lipid metabolism,	Disturbed lipid homeostasis [179]
26	JAZF1	Lipid/cholesterol metabolism and microglial efferocytosis [180]	Neuroinflammation by defective efferocytosis and defective lipid metabolism (not clear)
27	*SEC61G*	Protein trafficking, ER calcium leak channel [181]	*-*
28	*EPDR1*	Neurogenesis and synaptic signaling [162]	Not clear
29	*SPDYE3*	Cell cycle regulator [182]	-
30	*EPHA1*	Immune response, cholesterol metabolism, and synaptic function	Spine morphology abnormalities and synaptic dysfunction [183]
31	*CTSB*	Regulates apoptosis, neuroinflammation, and autophagy	lysosomal leakage of cathepsin B to the cytosol leads to neurodegeneration and behavioral deficits [184]
32	*SHARPIN*	Inflammation and immune system activation Synaptic signaling	Attenuated inflammatory/immune response [185]
33	*PTK2B*	Ca^2+^-activated non-receptor tyrosine kinase, involved in synaptic plasticity	Neuronal hyperexcitability by neuronal differentiation and electrical maturation [186]
34	*CLU*	Secreted by glia binds to Aβ and plays a protective role by preventing Aβ aggregation	Aβ clearance [187]
35	*ABCA1*	Cholesterol mobilization	Defective lipid metabolism, and neuroinflammation [188]
36	*ANK3*	Scaffolding proteins recruit diverse membrane proteins, (ion channels and cell adhesion molecules) into subcellular membrane domains	Altered neuronal excitability and altered neuronal connectivity [138]
37	*TSPAN14*	Regulates maturation and trafficking of the transmembrane metalloprotease ADAM10 [189]	Not clear
38	*BLNK*	Participates in the regulation of PLC-γ activity and the activation of Ras pathway [190]	Not clear. Possibly involved in immune regulation
39	*PLEKHA1*	Adaptive immunity	Inflammatory responses [191]
40	*USP6NL*	GTPase-activating protein involved in control of endocytosis	Dysfunction of the myeloid endolysosomal system [192]
41	*SPI1*	Controls microglial development and function	Regulating neuroinflammation [193]
42	*EED*	Catalyzes the methylation of histone and mediates the repressive chromatin	Synaptic dysfunction due to upregulation of synapse related gene [194]
43	*SORL1*	Regulates the recycling of the APP out of the endosome	Endosomal swelling and APP misprocessing [195]
44	*TPCN1*	Encodes a voltage-dependent calcium channel and involved in long-term potentiation in hippocampal neurons,	Altered calcium signaling and cognitive dysfunction [196]
45	*IGH* gene cluster	Immune response [197]	Not clear
46	*FERMT2*	APP metabolism and axonal growth	Impaired synaptic connectivity, and long-term potentiation in an APP-dependent manner [198]
47	*SLC24A4*	Neural development and cholesterol metabolism	Increased deposition of Ab and tau [199]
48	*SPPL2A*	Engaged in the function of B-cells and dendritic cells.	Activates TNF-α signaling [156]
49	*MINDY2*	Deubiquitination	Not clear
50	*APH1B*	γ-secretase	Brain atrophy and amyloid-β deposition [200]
51	*SNX1*	Endosome trafficking	Prevents BACE1 trafficking to the lysosomal degradation system, resulting in increased production of Aβ [201]
52	*CTSH*	Immune regulation	Role in neuroinflammation and amyloid β production [202]
53	*BCKDK*	Regulation of neurotransmitter synthesis, and mTOR activity.	Causes hyperexcitability, neuroinflammation, and dysregulation of neurotrophic factors [203]
54	*IL34*	Stimulates proliferation of monocytes and macrophages	Triggers neuroinflammation via colony-stimulating factor-1 receptor (CSF-1r) [204].
55	*PLCG2*	Present on microglia and function as immune regulator	Upregulated and activates inflammation related pathway [205]
56	*DOC2A*	A calcium sensor, facilitates neurotranbsmitter release in Ca^2+^-dependent manner	Abnormality in synaptic transmission [206]
57	*MAF*	Regulates T-cell susceptibility to apoptosis	Probably immune cell dysfunction and neuroinflammation [207]
58	*FOXF1*	Cell proliferation, cell cycle, and regulatory protein	Activated by PI3K/AKT and stress response and may cause inflammation [208]
59	*PRDM7*	Methyltransferases induce trimethylation	Possibly suppresses the synaptic gene [209]
60	*WDR81*	Facilitates the recruitment of autophagic protein aggregates and promotes autophagic clearance	Impaired autophagy [210]
61	*MYO15A*	A myosin involved in actin organization	Retromer dysfunction [211]
62	*GRN*	Regulates lysosomal biogenesis, inflammation, repair, stress response, and aging.	Neuroinflammation [211]
63	*SCIMP*	Immune regulation via major histocompatibility complex class II signaling.	Neuroinflammation [212]
64	*WNT3*	Synaptic function and immune regulation	Causes synaptic dysfunction and inflammation via Wnt3/β-catenin/GSK3β signaling pathway [213]
65	*ABI3*	Regulator of microglia	Neuroinflammation [214]
66	*TSPOAP1*	TSPO-associated protein 1, interacts with translocator protein (TSPO) and act indirectly to activate microglia	Neuroinflammation [215]
67	*ACE*	An endopeptidase	ACE has been shown to cleave amyloid-β (Aβ) [216]
68	*KLF16*	Regulates dopamine receptors	Modulates dopaminergic transmission in the brain [217]
69	*SIGLEC11*	Immune regulation	Proinflammation and phagocytosis [218]
70	*LILRB2*	Aβ receptor	Perturbance in synaptic signaling and cognitive impairment [219]
71	*ABCA7*	Lipid homeostasis and phagocytosis.	Disturbed lipid metabolism, ER stress. Impaired microglial response to inflammation
72	*RBCK1*	Involved in ubiquitination	TNF-α-mediated activation of NF-κB pathway.
73	*SLC2A4RG*	Encodes solute carrier protein. Involved in cell cycle via CDK1 pathway [220]	Not clear. Possibly induces proliferation
74	*CASS4*	Role in inflammation, calcium signaling, and microtubule stabilization.	Disturbed synaptic signaling and neuroinflammation [221]
75	*APP*	Proliferation, differentiation, and maturation of neural stem cells.	Abnormal cleavage causes plaque deposition [222]
76	*ADAMTS1*	APP hydrolysis	Increased Aβ generation through β-secretase-mediated cleavage [223]

**Table 4 ijms-25-02320-t004:** Risk loci identified in cross-ancestry genome-wide association study (GWAS), the associated function and disease mechanism in Parkinson’s disease.

S.No.	Gene (PD)	Function	Mechanism
1	*KRTCAP2*	Dementia-related gene	Inflammation and neurodegeneration [260]
2	*PMVK*	Involved in mevalonate pathway	Not clear. Possibly same as *GBA* [261]
3	*GBAP1*	Encodes for the enzyme glucocerebrosidase (GCase), a lysosomal enzyme	Lysosomal dysfunction [262]
4	*FCGR2A*	Phagocytosis and modulates inflammatory responses	Binds with IgG-specific immune complexes and activates signaling [263]
5	*VAMP4*	Endosomal trafficking of synaptic proteins	Impaired synaptic signaling and lysosomal degradation [264]
6	*NUCKS1*	Cell growth and proliferation [265]	Not clear
7	*RAB29*	Lysosome-related organelle biogenesis	Lysosomal dysfunction Axon termination [266]
8	*ITPKB*	Involved in inositol metabolism and calcium release from ER	Causes α-synuclein aggregation by dysregulated calcium release from ER-to-mitochondria [267]
9	*SIPA1L2*	Controls protein trafficking and BDNF/TrkB signaling [268]	Not clear; possibly abrupt synaptic signaling
10	*KCNS3*	Potassium channel	Neuroinflammation [269]
11	*KCNIP3*	Associated with inositol biosynthetic pathway [270]	Not clear
12	*MAP4K4*	Cell proliferation, inflammation, and stress response	Cytokine activation and neuroinflammation [271]
13	*TMEM163*	Influx or efflux transporter particularly Zn transport [272]	Not clear
14	*STK39*	Immune regulation	Inflammatory pathway [273]
15	*SATB1*	Transcriptional response in dopaminergic neurons	Senescence-mediated neuroinflammation [274]
16	*LINC00693*	Involved in miRNA processing complex [275]	Might affect protein expression and accumulation
17	*IP6K2*	Mitochondrial respiration	Mitophagy via PINK1 signaling [276]
18	*KPNA1*	Encodes importin α5 and is involved in lysosomal biogenesis and autophagy	Disturbed protein degradation [277]
19	*MED12L*	Transcriptional coactivation of nearly all RNA polymerase II-dependent genes, Wnt/beta-catenin pathway, and immune response [278]	Possibly transcriptional defects
20	*SPTSSB*	Regulates de novo synthesis of ceramides	Neuronal signaling, synaptic transmission, cell metabolism [279]
21	*MCCC1*	Mitochondrial enzyme and involved in leucine catabolism [280]	Possibly associated with mitochondrial dysfunction
22	*GAK*	Associated with lysosomal and chaperons	Defected lysosomal-mediated protein degradation [281]
23	*TMEM175*	Proton channel to maintain optimum pH in lysosomes	Downregulation of TMEM175 causes lysosomal over-acidification, impaired proteolytic activity, and facilitated a-synuclein aggregation [282]
24	*BST1*	Serves as a receptor that regulates leukocyte adhesion and migration	Immune regulation and inflammation [283]
25	*LCORL*	*-*	*-*
26	SCARB2	Encodes a receptor responsible for the transport of glucocerebrosidase (GCase) to the lysosome	Associated with lysosomal defects [284]
27	FAM47E	Present in close proximity to SCARB2	Lysosome/autophagy dysfunction
28	FAM47E-STBD1	-	-
29	SNCA	Dopamine release and transport, fibrillization of MAPT, and suppression of both p53 expression and transactivation of proapoptotic genes leading to decreased caspase-3 activation	Synuclein aggregation and induction of apoptosis [285]
30	*CAMK2D*	Calcium/calmodulin-dependent protein kinase ii delta. Involved in synaptic plasticity	Disturbed calcium signaling and synaptic function [286]
31	*CLCN3*	Ion channel transporter and neurotransmitter signaling [239].	Disturbed synaptic signaling
32	*ELOVL7*	Catalyzing the elongation of very long-chain fatty acids.	Possibly disturbed lipid metabolism and oxidative stress [287]
33	*PAM*	Glutamate receptor at parasynapses, associated with anxiety and hyperexcitation.	Disturbed glutamatergic and GABA signaling [288]
34	*C5orf24*	-	Function and mechanism are not clear. However, upregulated expression and DNA methylation in disease condition [289]
35	*LOC100131289*	-	-
36	*TRIM40*	E3 ubiquitin-protein ligase and inhibits NF-κB activity	Protein degradation and inflammation [290]
37	*HLA-DRB5*	Immune regulation	Inflammation [291]
38	*RIMS1*	Encodes a synaptic protein and involved in neurotransmitter release and synaptic transmission [277]	Possibly perturbance in synaptic signaling
39	*FYN*	Ion channel function, growth factor receptor signaling, immune system regulation.	Activates BDNF/TrkB, PKCδ, NF-κB, MAPK, Nrf2, and NMDAR signaling pathway and induces synuclein phosphorylation, inflammation and excitotoxicity [292]
40	*RPS12*	A special function in cell competition that defines the competitiveness of cells.	Not clear in PD; however, reported to cause inflammation [293]
41	*GPNMB*	Cell differentiation, migration, proliferation	Interacts with a-synuclein and induces its phosphorylation, cellular internalization, and fibrillization [294]
42	*GS1-124K5.11*	-	-
43	*CTSB*	Lysosomal hydrolase cathepsin B involved in waste degradation in cells	Autophagy and lysosomal dysfunction causing a-synuclein aggregation [295]
44	*FGF20*	Maintenance of dopaminergic neurons	Affects dopaminergic neurons in paracrine manner [296]
45	*BIN3*	Cytokinesis and RNA methyltransferase	Probably target transcription and translation step [297]
46	*FAM49B*	Regulates mitochondrial function	Mitochondrial fission, oxidative stress, and inflammation [298]
47	*SH3GL2*	Encodes Endophilin A which regulates autophagy in calcium dependent manner	Autophagy dysfunction at synapses [299]
48	*UBAP2*	Synapse formation, maintenance, and signaling	-
49	*ITGA8*	Alpha8 integrin, cell adhesion, cell signaling, and cytoskeletal organization	Increases cell-to-cell transfer of a-Syn [300]
50	*GBF1*	Maintenance and function of the Golgi apparatus, and mitochondria migration and positioning	Increase in Golgi fragmentation [301]
51	*BAG3*	A chaperone and regulates autophagy	Its downregulation promotes autophagy dysfunction and disease progression [302]
52	*INPP5F*	PI4P-phosphatase Involved in endocytic pathway	Disturbed endocytosis [303]
53	*RNF141*	-	-
54	*DLG2*	DLG2-encoded protein involved in glutamate receptor phosphorylation	Phosphorylation of NR2 subunit and hyperexcitability [280]
55	*IGSF9B*	Cell adhesion molecule at inhibitory synapses and plays role in neuroplasticity and synaptic transmission	Any disturbance in inhibitory synapses causes dysregulation of information flow and cognitive defects [304]
56	*LRRK2*	Associated with intracellular membranes and vesicular structures	Causes accumulation of a-synuclein, which activates MAPK signaling and microglial activation leading to inflammation [305]
57	*SCAF11*	Encodes a caspase	-
58	*HIP1R*	Clathrin-mediated endocytosis, actin dynamics, intrinsic cell death pathway [306]	Not clear. Probably affects the endocytosis of a-synuclein and activate caspase response
59	*FBRSL1*	-	-
60	*CAB39L*	Encodes calcium binding 39-like protein	-
61	*MBNL2*	Encodes for the muscleblind-like protein 2, which belongs to a conserved family of RNA-binding proteins	Reduced MBNL2 expression accompanied by the reduction in a developmental RNA processing [307]
62	*MIPOL1*	-	-
63	*GCH1*	Essential for dopamine production	Affects dopaminergic signaling [308]
64	*RPS6KL1*	-	-
65	*GALC*	Encodes galactocerebrosidase	Impaired autophagy and disturbed protein trafficking causes a-synuclein deposition [309]
66	*VPS13C*	Localized to the outer membrane of mitochondria	PINK1/Parkin-dependent mitophagy [310]
67	*SYT17*	Encodes synaptotagmin-17, associated with vesicle trafficking and transport at synapses	Disturbed synaptic trafficking [311]
68	*CD19*	Immune regulatory molecule presents on B lymphocyte	Neuroinflammation by suppressing local immune response [312]
69	*SETD1A*	Histone methyltransferase	Might affect synaptic signaling, excitation and glutamatergic signaling [313]
70	*NOD2*	Immune homeostasis	Nox-2-mediated oxidative stress and neuroinflammation followed by loss of dopaminergic neurons [314]
71	*CASC16*	-	-
72	*CHD9*	Activates transcription factor CREBPP, Involve in Notch signaling	Aberrant survival signaling pathway [315]
73	*CHRNB1*	Encodes subunit of the n-acetylcholine receptor, Ion channels, transporters, and neurotransmitter signaling [239]	Disturbed cholinergic signaling
74	*RETREG3*	Involved in ER autophagy	Activation of ER autophagy by mTOR signaling [316]
75	*UBTF*	Transcription factor associated with ds-DNA break and apoptosis	Altered protein expression [317]
76	*BRIP1*	Facilitates repair of SSBs and DSBs	Excitotoxicity, mitochondrial damage, and cell death [318]
77	*DNAH17*	Encodes dynein axonemal heavy chain 17 involved in cytokinesis, microtubule-based movement, mitotic spindle organization, meiotic nuclear division [319]	-
78	*ASXL3*	-	-
79	*RIT2*	Involved in lysosomal activity	Activate *LRRK2* gene and lysosomal dysfunction and leads to a-synuclein deposition [320]
80	*MEX3C*	Encodes RNF 194, an RNA binding protein impart immunoregulatory role	Neuroinflammation [321]
81	*SPPL2B*	-	-
82	*CRLS1*	Encodes cardiolipin synthase 1, involved in mitochondrial membrane formation	Mitophagy [322]
83	*DYRK1A*	Synaptic and nuclear proteins, including transcription factors	Causes phosphorylation of a-synuclein and downregulates PI3K/AKT pathway to induce apoptosis [323]
84	*FAM171A2*	Downstream of GRN, is a novel genetic regulator of progranulin production expressed on microglial surface	Downregulates progranulin level in brain [324]
85	*CRHR1*	Encodes corticotropin-releasing hormone receptor, involved in regulation of stress and immune responses	Downregulates CREB signaling [325]
86	*WNT3*	Immune regulation	PD-related gene expression in immune cells [291]

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
