# Peer review of "Unraveling the Genetic Landscape of Neurological Disorders: Insights into Pathogenesis, Techniques for Variant Identification, and Therapeutic Approaches"

_ijms, 2024, doi:10.3390/ijms25042320_

Round 1

Reviewer 1 Report

Comments and Suggestions for Authors

This is an excellent review article about genetic influences on common neurodegenerative diseases (NDD). The authors cover known familial gene mutations, and most importantly, introduce the concept of risk genes compiled from genome-wide association studies (GWAS). As the authors also point out, GWAS studies define  minority of risk factors for NDD's, and therefore there is much yet to learn. The authors finish with a nice introduction to available gene therapy approaches and present several gene therapies currently in research studies (or planned for research studies).

I support publication of this review article after attention to a very few minor corrections needed. On pg 177 "168" should be "1.68".

Comments on the Quality of English Language

There are a very few minor subject-verb anomalies that would be corrected with careful editing.

Author Response

Thanks for the comments. We have modified the manuscript and corrected the errors suggested.

Reviewer 2 Report

Comments and Suggestions for Authors

My suggestions:

1. Figure 1 needed to be uploaded in better resolution, it is difficult to read/see.

2. In Chapter 2 I would add a table, which compares the genetic technologies, their basics, benefits, and disadvantages.

3. Besides ALS, I would also mention frontotemporal dementia (FTD). ALS and FTD have several common genetic factors (C9orf72, TARDBP, FUS etc)

4. Among AD-related genes I would also mention ABCA7, since its role in EOAD may also be possible.

5. I would add a shorter chapter, which briefly discusses the possible monogenic neurodegenerative diseases, such as prion disease, Huntington's disease, or multiple sclerosis. 

6. The authors may mention some successful approaches in humans for gene therapies in other diseases, including SMA. 

7. Table 1-2-3 are too basic. I would add additional information to the genetic factors, including functions, possible disease-related mechanisms, and examples of genetic mutations. It may be better to add these tables into a supplemnet file. 

Author Response

Reviewer 2

1. Figure 1 needed to be uploaded in better resolution, it is difficult to read/see.

Response: Thanks for the comments. We uploaded a high-resolution figure as suggested.

2. In Chapter 2 I would add a table, which compares the genetic technologies, their basics, benefits, and disadvantages.

Response: Thanks for the comments. We added a table as suggested.

3. Besides ALS, I would also mention frontotemporal dementia (FTD). ALS and FTD have several common genetic factors (C9orf72, TARDBP, FUS etc)

Response: Thanks for the comments. We have revised the manuscript and included FTD as suggested.

4. Among AD-related genes I would also mention ABCA7, since its role in EOAD may also be possible.

Response: Thanks for the comments. We have included ABCA7 as suggested.

5. I would add a shorter chapter, which briefly discusses the possible monogenic neurodegenerative diseases, such as prion disease, Huntington's disease, or multiple sclerosis. 

Response: Thanks for the comments. We have revised the manuscript as suggested.

6. The authors may mention some successful approaches in humans for gene therapies in other diseases, including SMA. 

Response: Thanks for the comments. We have revised the manuscript as suggested.

7. Table 1-2-3 are too basic. I would add additional information to the genetic factors, including functions, possible disease-related mechanisms, and examples of genetic mutations. It may be better to add these tables into a supplemental file. 

Response: Thanks for the comments. We have modified the Tables as suggested.

Reviewer 3 Report

Comments and Suggestions for Authors

This review article offers and in depth analysis of the genetic landscape of the neurodegenerative disorders, with a primary focus on amyotrophic lateral sclerosis (AML), Alzheimer Disease (AD) and Parkinson’s Disease (PD).

These diffuse and devastating pathologies result from a not well-understood interaction of genetic, environmental and familial factors. 

A notable strength of this work is to comprehensively highlight the vast complexity of these disorders in terms of onset, diagnosis, penetrance of symptoms and inheritance and to dissects the genetic changes found in ALS, AD and PD.  Indeed, the application of cutting-edge high-throughput technologies and long read whole-genome sequencing has greatly favored the identification of genetic variations in neurodegenerative disorders, addressing the complexity and remarkable heterogeneity of these pathologies and their heritability.

The review comprehensively summarizes these findings, the progress into the comprehension of the molecular pathways involved and the pathogenic mechanisms as well as the difficulties in unraveling strong causative association of the discovered genetic variations and candidate genes with the neurodegenerative diseases.

There is currently no treatment that can cure ALS, AD and PD or halt the clinical progression of these diseases. There is, therefore, a need for new treatment interventions.  

In this perspective, the review dedicates relevant space to the discussion of approaches of precision medicine and in particular of gene therapy and gene editing strategies. These new therapeutic techniques are potential game-changers in the clinical management of neurodegenerative diseases and the paper appropriately highlights the recent breakthrough of gene transfer and precise gene correction, including the description of some clinical trials. At the same time, the recognition of safety issues and the future challenge of improving the effectiveness and specificity of vector technology, reflects the choice to give a realistic and critical approach to the discussion.

In summary, the present review offers a valuable overview of the current advancements and limitations of our comprehension of the major neurodegenerative disorders, including the perspectives of new therapeutic approaches. This work is not redundant with respect to other published reviews on this subject and may add a commendable contribution for researchers involved in the field.

The style of the manuscript is neat and the reading flows smoothly. 

I detected an error in the figure 5 (top left): is Design of gRNA instead of Design of pegRNA.

Comments on the Quality of English Language

Despite a few corrections needed and typos, the English is of good quality.

Author Response

Reviewer 3

This review article offers and in-depth analysis of the genetic landscape of the neurodegenerative disorders, with a primary focus on amyotrophic lateral sclerosis (AML), Alzheimer Disease (AD) and Parkinson’s Disease (PD). These diffuse and devastating pathologies result from a not well-understood interaction of genetic, environmental and familial factors. 

A notable strength of this work is to comprehensively highlight the vast complexity of these disorders in terms of onset, diagnosis, penetrance of symptoms and inheritance and to dissects the genetic changes found in ALS, AD and PD.  Indeed, the application of cutting-edge high-throughput technologies and long read whole-genome sequencing has greatly favored the identification of genetic variations in neurodegenerative disorders, addressing the complexity and remarkable heterogeneity of these pathologies and their heritability.

The review comprehensively summarizes these findings, the progress into the comprehension of the molecular pathways involved and the pathogenic mechanisms as well as the difficulties in unraveling strong causative association of the discovered genetic variations and candidate genes with the neurodegenerative diseases.

There is currently no treatment that can cure ALS, AD and PD or halt the clinical progression of these diseases. There is, therefore, a need for new treatment interventions.  

In this perspective, the review dedicates relevant space to the discussion of approaches of precision medicine and in particular of gene therapy and gene editing strategies. These new therapeutic techniques are potential game-changers in the clinical management of neurodegenerative diseases and the paper appropriately highlights the recent breakthrough of gene transfer and precise gene correction, including the description of some clinical trials. At the same time, the recognition of safety issues and the future challenge of improving the effectiveness and specificity of vector technology, reflects the choice to give a realistic and critical approach to the discussion.

In summary, the present review offers a valuable overview of the current advancements and limitations of our comprehension of the major neurodegenerative disorders, including the perspectives of new therapeutic approaches. This work is not redundant with respect to other published reviews on this subject and may add a commendable contribution for researchers involved in the field.

The style of the manuscript is neat and the reading flows smoothly. 

I detected an error in the figure 5 (top left): is Design of gRNA instead of Design of pegRNA.

Despite a few corrections needed and typos, the English is of good quality.

Response: Thanks for the comments. We have modified the figure and corrected the errors suggested.

Round 2

Reviewer 2 Report

Comments and Suggestions for Authors

Authors fulfilled my suggestions.